# Influence Scores at Scale for Efficient Language Data Sampling

**Nikhil Anand**[*] and  **Joshua Tan**[*] and  **Maria Minakova**
Amazon Alexa AI

## Abstract

Modern ML systems ingest data aggregated from diverse sources, such as synthetic, human-annotated, and live customer traffic. Understanding *which* examples are important to the performance of a learning algorithm is crucial for efficient model training. Recently, a growing body of literature has given rise to various "influence scores," which use training artifacts such as model confidence or checkpointed gradients to identify important subsets of data. However, these methods have primarily been developed in computer vision settings, and it remains unclear how well they generalize to language-based tasks using pretrained models.

In this paper, we explore the applicability of influence scores in language classification tasks. We evaluate a diverse subset of these scores on the SNLI dataset by quantifying accuracy changes in response to pruning training data through random and influence-score-based sampling. We then stress-test one of the scores – "variance of gradients" (VoG) from Agarwal and Hooker (2022) – in an NLU model stack that was exposed to dynamic user speech patterns in a voice assistant type of setting. Our experiments demonstrate that in many cases, encoder-based language models can be fine-tuned on roughly 50% of the original data without degradation in performance metrics. Along the way, we summarize lessons learned from applying out-of-the-box implementations of influence scores, quantify the effects of noisy and class-imbalanced data, and offer recommendations on score-based sampling for better accuracy and training efficiency.

## 1  Introduction

A salient challenge in training transformer-based models is selecting *which* examples are most important for learning. Understanding the relative importance of training examples towards model performance can inform data selection strategies that minimize customer privacy risks associated with the collection of training data, estimate the impact of the removal of copyrighted or sensitive data, determine mixing strategies to augment monolingual and multilingual datasets to improve accuracy, and identify defective subsets of data. At the same time, in cases where it is desirable to train on as much data as possible – such as large language models – determining the influence of different data instances (both contextually and during pre-training) can help identify failure modes at the level of specific tokens (Grosse et al., 2023), determine the impact of removal of intellectual property, and significantly reduce costs through more efficient model training (Renduchintala et al., 2023).

A growing body of literature in the science of deep learning aims to capture this hierarchy of example importance and has led to a proliferation of a number of "difficulty" or "influence" scores (e.g., Paul et al. (2021); Agarwal and Hooker (2022); Toneva et al. (2019); Ethayarajh et al. (2022); Garima et al. (2020); Sorscher et al. (2022); Swayamdipta et al. (2020); see App. A for a more complete review). These scores use various training artifacts, such as the margin of confidence or the variance of loss gradients, to rank the relative contribution of each example to model performance. This ranking of examples can then be used in many downstream tasks that require intelligent data selection, such as pruning datasets while maintaining or even improving model accuracy (Paul et al., 2021; Sorscher et al., 2022; Marion et al., 2023); identifying outliers and misannotations in labeled data (Garima et al., 2020; Ethayarajh et al., 2022; Pleiss et al., 2020a; Carlini et al., 2019; Feldman and Zhang, 2020); or reweighting/reordering training examples to increase model robustness (Ren et al., 2018; Wu et al., 2021).

Apart from a few notable exceptions (Swayamdipta et al., 2020; Ethayarajh et al.,

---
[*]These authors contributed equally, correspondence: nkhlanan@amazon.com, jshtan@amazon.com.

2022; Marion et al., 2023), influence scores have primarily been developed and demonstrated in the context of image classification, and relatively little is known about their efficacy in downstream language-based tasks.[1] The application of these scores to data selection is further complicated by the fact that during fine-tuning, modern ML systems often ingest a vast amount of data that come from multiple sources, such as synthetic, weak signal,[2] live customer data, and human-annotated. Beyond quantifying the efficacy of influence scores in this highly mixed data setting, there is an operational question of the *existence* of a simple, scalable influence score that can be easily accommodated in a production workflow.

In this work, we take a first pass at answering these questions. First, we benchmark a subset of influence scores on the SNLI dataset (Bowman et al., 2015) in the downstream task of data reduction using a pretrained BERT model (Devlin et al., 2019). Given the task of pruning a language dataset for fine-tuning, are influence scores useful signals for determining optimal data selection strategies? If so, which scores work best? We evaluate these scores against a random sampling baseline, in both noisy and clean data settings.

User speech patterns are constantly evolving due to current events as well as user-system interactions that can be difficult to anticipate. Are influence scores still effective in surfacing data critical for model performance in this dynamic setting? To answer this question, we build upon on our initial findings on SNLI and implement one influence score ("variance of gradients" or "VoG", first presented in Agarwal et al. (2022)) in a generic, large-scale NLU model stack commonly found in commercial voice assistants. We present results for existing in-house test data as well as results for a live user study in which we leveraged VoG scores for the purpose of substantially reducing training data without incurring model-performance degradation.

Among the five influence scores we evaluated on SNLI, most out-of-the-box implementations do not beat a baseline of randomly pruning the dataset. The implementations can be improved to do better than the random-pruning baseline, but this typically

requires careful experimentation to tune hyperparameters specific to each score. Out of the scores we tested, we find that VoG performs best relative to the random-pruning baseline, particularly at large pruning fractions. Test accuracy is mostly maintained after pruning $\sim$45% of the SNLI training data using VoG scores calculated in a "one-shot" fashion, i.e. from a single training run, without any score hyperparameter tuning.

In a large-scale user study performed using the NLU stack, we find that sampling by VoG scores is effective at surfacing training data that is particularly efficient for learning. We prune roughly 50% of training data without incurring statistically significant regressions in key metrics that track NLU errors, *relative to a baseline model trained with all data*.

## 2 Experiments on SNLI

### 2.1 Selection of influence scores

We considered five different influence scores (described in Table 1) to benchmark in data-reduction tasks on SNLI (Bowman et al., 2015), based on the following criteria: first, they should not require extensive computational resources to implement. For example, the score should not require extensive ensemble averaging by training many ($\gg 1$) copies of "replicate" models to refine the influence measurement of any particular example since many production models can only be trained once in operational workflows.[3] Second, the scores should have a natural definition in language models. This excluded some scores that were originally defined in the context of computer vision, such as input-pixel perturbation (Su et al., 2019). We report the implementation details of these scores in App. B.1. Our experiments on SNLI are run on BERT$_{\text{SMALL}}$ ($L = 4$, $H = 512$, 29.1M parameters), but we comment on the effects of model size in App. B.3.

### 2.2 Experimental Setup

We ran two sets of data-pruning experiments on SNLI to understand the effectiveness of pruning based on the influence scores in Table 1.

In Section 2.3, we describe data-pruning experiments on the original SNLI dataset. First, we generated the influence scores in Table 1 for the entire

---

[1] Large language models such as GPT-3 and T5 do implement some basic data mixing strategies (Brown et al., 2020; Raffel et al., 2020). Our focus here, however, is the setting of using a pretrained model in a downstream task.

[2] For example, data not associated with customer interruptions in online traffic, which are then pseudo-annotated with labels according to the top model hypothesis.

[3] We report on the results of scores that require a moderate $\mathcal{O}(1)$ number of re-runs such as EL2N (Paul et al., 2021), but our main motivation is to determine if there are influence scores that can be used in a "one-shot" setting, using only training artifacts generated from a single run.

| Score | Description |
|---|---|
| VoG (Agarwal and Hooker, 2022) | Variance of gradients of model outputs with respect to the inputs. |
| EL2N (Paul et al., 2021) | Norm of the margin of confidence between the model prediction and the one-hot label. |
| Forgetting Score (Toneva et al., 2019) | How often an example is forgotten i.e. goes from being classified correctly at checkpoint $i$ to incorrectly at $i+1$. |
| PVI (Ethayarajh et al., 2022) | Fine-grained information-theoretic quantity whose expectation value is the amount of usable information (in bits) by the model. |
| TracIn (Garima et al., 2020) | Influence of any example $z$ towards another example $z'$ by tracking their gradient dot products. We generate the self-influence scores where $z = z'$. |

Table 1: Description of the influence scores we generated on the SNLI dataset. See App. B.1 for implementation details.

SNLI training data. We then pruned the training data by using the scores to sample either "easy" or "hard" examples,[4] and measured test accuracy for a model trained on the reduced dataset. We compared these score-sampling pruning results to pruning by random sampling. We defer details of the implementation of influence scores and additional findings to App. B.1, but note here that we consider two normalization schemes for VoG scores: **class-normalization**[5], where the scores are normalized with respect to the mean and standard deviation of each class, and **dataset-normalization**, with respect to the full training dataset.

Results on a relatively clean public dataset like SNLI may not always translate to results on large, commercial datasets that are noisier and highly class-imbalanced. In Section 2.4, we address this concern by running similar pruning experiments on SNLI with increasing levels of randomly generated label noise.[6] We then computed VoG, TracIn, and PVI scores[7], pruned easy/hard examples based on those scores, and compared test accuracy to a random sampling baseline.

---

[4]We use the terminology "easy" and "hard" for pedagogical reasons. Strictly speaking, we are running data-pruning experiments where examples are sampled from either the head or tail of the score distributions. Frequently, these examples do correspond to what a human would find easy and hard, respectively, but we clarify in Section 2.3 when they do not.

[5]As originally prescribed in Agarwal and Hooker (2022).

[6]Details about the label noise are given in App. B.6.

[7]These scores were chosen in the noisy label setting due to their reported efficacy in surfacing defective data.

## 2.3 Results on SNLI

Fig. 1 shows test accuracy at the end of training as a function of percent data pruned for each of the five score-based pruning strategies.

**General Findings**: For most scores, we found that pruning the hardest examples resulted in models with poorer test accuracy compared to pruning the easiest examples. This supports the findings of Sorscher et al. (2022), which hypothesized that hard examples contain critical information about the decision boundaries of classes in larger, less noisy datasets. We also find that out-of-the-box implementations of influence scores – with the exception of VoG – do not result in test accuracy higher than the random sampling baseline without score hyperparameter tuning. For example, for EL2N scores, it is crucial that the scores are computed early during fine-tuning for best results. We explored different implementations and chose those that gave best results for data pruning, while adhering to the criteria listed in Sec. 2.1.

**VoG**: Remarkably, VoG required only a single model training run and no hyperparameter tuning. At 45% of training data removed, pruning class-normalized VoG-easy examples led to a test accuracy of 85.04±0.20%, compared to 85.52±0.14% with all of the data. At smaller pruning fractions ($\lesssim$10%), performance is roughly within the margin of error of sampling randomly. We find that sampling dataset-normalized scores generally performs worse than class-normalized (84.60±1.50% at 45% easy pruned), which is due to the over-representation of the "contradiction" class (Fig. 2) in the tail. We will revisit the merits of class versus dataset normalization in Sec 3.

**EL2N**: Best results were obtained by computing EL2N scores early in training; we found epoch $\sim$ 2 outperformed the random pruning baseline for small to moderate pruning fractions (between 0-25%), but worse beyond that.

EL2N is a margin-based metric, which means that examples drawn from the *tail* of the EL2N distribution should lie close to the decision boundary between classes (Paul et al., 2021; Sorscher et al., 2022). If that is true, then removing these examples should dissolve the decision boundary between different classes, and account for the drop in test accuracy. We provide some evidence for this in App. B.4 by clustering the t-SNE (Hinton and Roweis, 2002) encoder representations of the training and test data, before and after pruning EL2N-

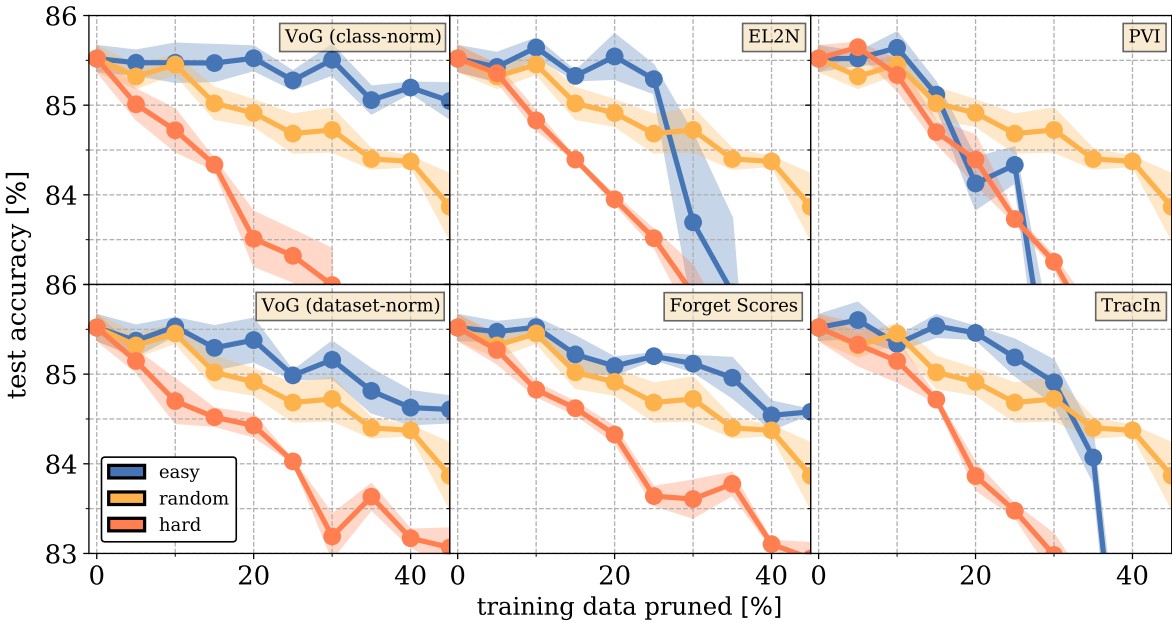

Figure 1: Shows the test accuracy on SNLI as a function of training data pruned for different influence scores (pruning easy examples is blue, random in gold, and hard in orange). Points show the mean test accuracy computed over three training runs, and shaded regions indicate the 1-$\sigma$ envelope.

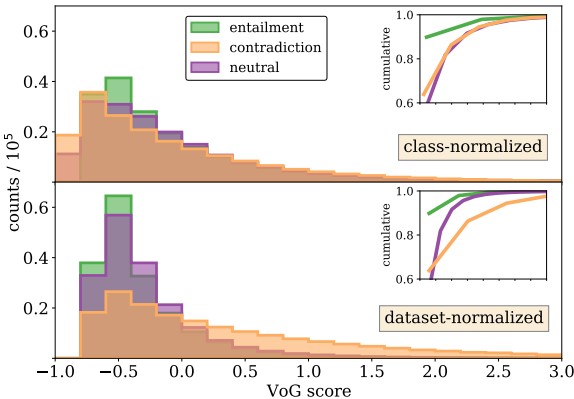

Figure 2: Distribution of VoG scores for SNLI dataset with two different normalization schemes.

hard train data.

**PVI**: We found that beyond a small fraction (5-10%) of data pruned, pruning by PVI scores generally did not outperform random pruning.[8] Although manual inspection of the top negative-scoring PVI examples showed that this score was effective at finding several misannotated examples in SNLI (see App. B.5), the number of such misannotations was quite small, and beyond a certain pruning fraction, the test accuracy fell off rapidly[9].

---

**Forgetting Scores**: We observe consistent improvements over the random sampling baseline when pruning the least forgotten examples, with a test accuracy of 84.58±0.02% at 45% data pruned. However, due to the rapid convergence of fine-tuning (resulting in most examples having zero forgetting score), forgetting events for the entire training set had to be logged at a high cadence (once every 50 training steps), making it challenging to apply in a production setting.

**TracIn**: Pruning up to $\sim 30\%$ of training data by TracIn scores led to consistent improvement over training on randomly pruned data. Similar to pruning EL2N-hard examples, pruning TracIn-hard examples dissolves the decision boundary between classes. The similarity index[10] of the top 5% of hard examples for these two scores is 0.37 (versus 0.11 for random sampling), indicating they are roughly sampling the same difficult examples.

### 2.4 Results on SNLI with Added Label Noise

Fig. 3 shows the results of our noisy data reduction experiment, where the amount of isotropic label noise was varied from five to 30 percent. We observed that pruning VoG-easy examples outperformed the random-pruning baseline in all of the

---

[8]Sampling examples from the head of the PVI score distribution corresponds to "hard" or potentially misannotated examples, while the tail corresponds to "easier" examples.

[9]While outside the scope of this work, the explicit dependence on the model inductive bias through the "null model" in the definition of PVI suggests that it may be more effective at

iterative pruning, rather than single-shot pruning where scores are computed only once for the model trained on all data.

[10]Given by the Jaccard index for two sets $A$ and $B$: $\frac{|A \cap B|}{|A \cup B|}$.

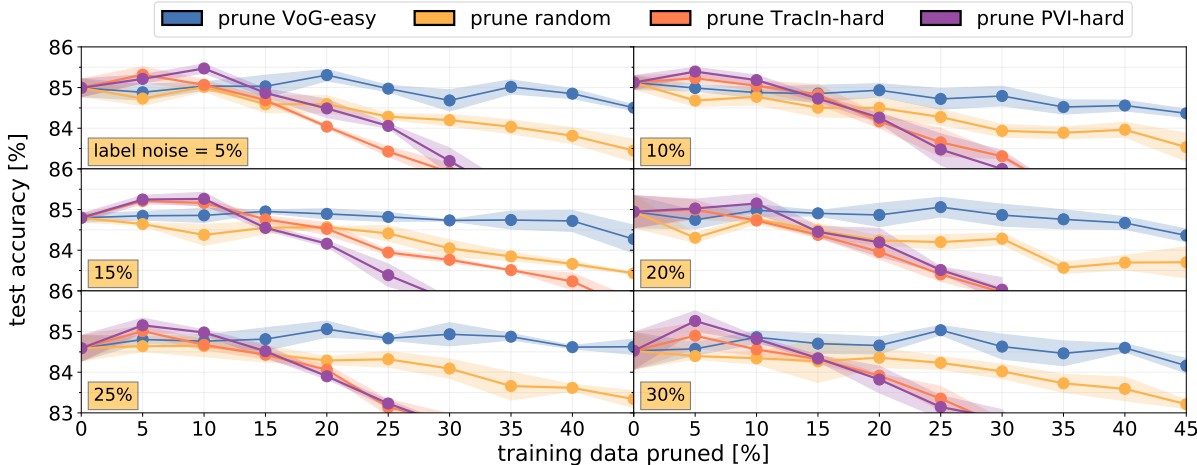

Figure 3: Shows the results of pruning VoG-easy (blue), TracIn-hard (orange), PVI-hard (purple), or randomly selected (gold) examples for noisy SNLI. The amount of added isotropic label noise is given in the orange inset. Points indicate the average of three runs and shaded regions indicate the 1-$\sigma$ envelope.

noisy settings, for large pruning fractions. In some cases, pruning based on VoG scores even clawed back a fraction of the initial accuracy loss due to noisy labels. However, somewhat surprisingly, this was likely *not* because VoG-based selection was pruning the misannotated examples themselves. The similarity index between the easiest VoG examples and all of the introduced misannotated examples in the 30% label-noise setting was only $\approx 0.11$. Compared to random sampling ($\approx 0.11$), we conclude that VoG does not do better than random chance at finding misannotated SNLI examples, but *does* reliably extract more influential examples that partially mitigate the effects of label noise. In all noisy settings, we found that pruning VoG-hard examples did worse than pruning randomly.

Pruning by TracIn and PVI scores resulted in a small but persistent accuracy increase when $\sim$ 5-10% of the hard data was pruned. In the 30% noise setting, the similarity index between the TracIn-hard and PVI-hard examples and misannotated examples was $\approx 0.11, 0.09$, respectively, again indicating that the accuracy gains are not due to the removal of defects. The number of instances with a PVI score of $< 0$ (indicating a potential misannotation) comprises only 6% of the mislabeled data. Nevertheless, it appears beneficial to use these scores to prune 5-10% of overall hard data that adversely impacts training in a noisy setting.

## 3   VoG in the Context of NLU

Given its promising results for data reduction on SNLI, we set out to evaluate VoG in an environment typically found in large, general-purpose commercial voice assistants. This setting poses practical challenges often not reflected in public datasets, such as noisy and evolving speech patterns, diverse vocabularies, dialects, carrier phrases, and out-of-distribution named entities. As an added challenge, we focus on Japanese-data trained models and datasets to determine if VoG-based influence scoring could function in a lower-resource setting. The statistical-model component in this setting consisted of generic domain classifier (DC), intent classifier (IC), and named entity recognition (NER) models organized in a coupled, hierarchical manner: a single multi-class DC model is trained on all domains' data to first predict the domain for a given utterance, which then invokes a domain-specific joint IC-NER model trained on in-domain data[11]. Both sets of models were based on distilled Japanese BERT models and VoG scores were computed using the procedure given in App. B.1.

Fig. 4 shows the class-normalized scores for a subset of five domains that differ in the proportion of the overall training data they represent and intent-label complexity[12] within that domain. We observe that smaller domains tend to have higher scores (e.g., HealthAndFitness vs. HomeAutomation) and more complex domains tend to have higher scores (Shopping vs. Video). In some cases, domains that are similar in size and complexity still exhibit different VoG score distributions (Music vs. Video) which reflects differing influence of domain data

---
[11]See App. C.1 for additional context.
[12]As measured by Shannon entropy; see App. C.6.

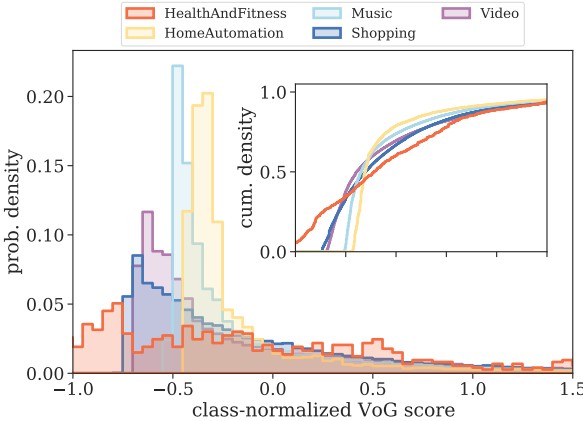

Figure 4: Probability density of class-normalized VoG scores for data used to train in-house models, grouped by ground-truth domain.

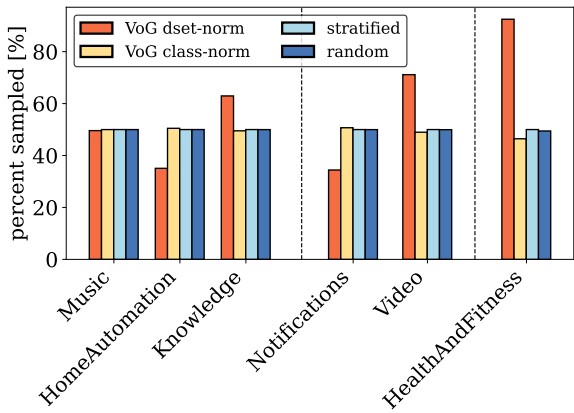

Figure 5: Reduction of training data using different sampling techniques for three large domains (left), two medium domains (middle), and a small domain (right).

due to factors that cannot be easily discerned without extensive manual analysis.[13]

### 3.1 Experiment Setup

Here we describe in-house experiments which leverage VoG scores for frugal data selection. We present evaluation results on both existing internal data and live user interaction data to determine the impact of pruning training data. In both sets of experiments, models were trained and evaluated on de-identified, historical user data.

**Sampling Technique:** We benchmarked VoG-based data sampling against random sampling and stratified sampling.[14] In stratified sampling, we sample utterances randomly *while preserving the domain distribution* of the training data. Fig. 5 shows the relative reduction of a few domains' training data when pruning using different sampling techniques. Sampling by dataset-normalized VoG scores led to highly non-uniform reductions, as expected since those scores reflect data influence with respect to all domains' training data.

For VoG-based sampling, we used a probabilistic method where the sampling likelihood was proportional to the score (see App. C.3 for details). This results in higher likelihood of pruning training data with scores located near the head (low-score portion) of the VoG distribution. We computed scores using training checkpoints of a baseline model and used those scores as sampling weights to create a new pruned training dataset used for the candidate model.

---

[13]We include additional domain-level analysis in App. C.4.

[14]In each case, sampling was performed without replacement.

**Experiments on De-identified Historical Data:** We compared the performance of models trained on the complete de-identified historical training data versus on a reduced subset of it. The sampled data was used for fine-tuning of the DC model and the in-domain subset of that sample was used for fine-tuning IC-NER stat models. Each model was trained for the same number of epochs without early stopping.

Due to the non-deterministic nature of probabilistic sampling, we averaged over three random seeds for each sampling technique, trained models on those samples, and report evaluation results.

The majority of experiments investigated model performance and efficiency in the context of aggressive reduction of training data (roughly 50%). We performed additional experiments on VoG-based sampling techniques in which we pruned roughly 14% of historical data, in order to understand the impact on model performance when targeting less aggressive data-reduction targets.

**User Study:** In a randomized user study, we investigated the impact of pruning roughly half of the training data. The scaled user study exposes the model to unconstrained human speech, which varies (often dramatically) in carrier phrase frequency, vocabulary, and named entities distribution compared to public datasets, offering a challenging setting to evaluate the efficacy of VoG-based scoring. To ensure that the most frequent user requests are captured, large-scale NLU systems often consist of both statistical models as well as deterministic model artifacts. In addition, although these statistical models are primarily trained on historical data, they also are trained on additional

in-house synthetic data. Thus, in order to understand how our results might generalize to a production system, we compared composite NLU models containing both statistical and deterministic components, trained on both historical and synthetic data. The *total* training-data size reduction for the candidate model versus the baseline model was approximately 40%.

The user study was performed using an in-house experimentation platform that provided signals of model performance that captured potential degradations to the user experience. Pruning solely by DC VoG scores led to some downstream NER-related regressions for the composite model when evaluated on historical data. Therefore, as an extra precaution, we pruned training data based on a modified version of DC-model VoG scores. We first estimated NER complexity for each intent using the Shannon entropy of slot-label-trail annotations (i.e. all data labeled with a given intent were assigned the same complexity score). The final sampling scores were taken to be the mean of the DC VoG scores and estimated NER complexity scores. For details, see App. C.6.

The user study ran for 11 days, with users distributed equally between the baseline and candidate models.

## 3.2 Experiment Metrics

In experiments on historical data we measured performance on held-out test data[15] in terms of component-wise error rates. We measured domain and intent classification performance using the recall-based classification error rates DCER and ICER.

To evaluate slot-filling performance, we measured semantic error rate (SEMER):

$$\texttt{SEMER} \equiv \frac{\text{\# Intent errors} + \text{\# Slot errors}}{\text{\# Test data} + \text{\# Slots}}.$$

Specifically, we measured F-SEMER, the harmonic mean of SEMER using predicted labels as the reference and SEMER computed on ground-truth labels as the reference; this score balances precision/recall equally. We also report the interpretation error rate IRER, which reflects the rate of any kind of error (slots, intents, domain). For all error-rate metrics, we report the error rate of the model trained on

pruned data *relative* to the model trained on all data:

$$\texttt{Relative ER} \equiv \frac{\Delta\texttt{ER}}{\texttt{ER}_{\text{no-pruning}}}.$$

It is useful to define a metric that measures accuracy loss per utterance, relative to the no-pruning baseline. We report relative **data-score efficiency**, originally proposed by Çano and Bojar (2019):

$$\sigma_{\texttt{ER}} \equiv \frac{\Delta\texttt{ER}/\texttt{ER}_{\text{no-pruning}}}{\Delta d/d_{\text{no-pruning}}}, \tag{1}$$

where $\texttt{ER}_{\text{no-pruning}}$ and $d_{\text{no-pruning}}$ correspond to the error rate and number of training instances for the model trained on all data.[16] $\sigma$ values express the ratio of relative-change in performance to the relative-change in training data. In our data-pruning setting, $\Delta d$ is negative. More positive values of $\sigma_{\texttt{ER}}$ indicate less model degradation due to pruning, and a $\sigma_{\texttt{ER}}$ score of zero indicates no model-performance regression relative to the no-pruning baseline.

We analyzed model performance in the *user study* using two in-house proprietary metrics developed to detect model defects involving user requests:

**Predicted Defect Rate (PDR):** PDR is a model-based metric that uses both the system response and user-provided signals (e.g., whether they interrupted the device response) as features to produce a binary prediction for each request indicating whether that request likely resulted in a user-perceived defect. We also report *tail-PDR*, which is PDR corresponding to the bottom 40% of user traffic. These less common requests are much less likely to be covered by deterministic components.

**Unrecoverable Error Rate (UER):** This metric tracks cases where the utterance cannot be acted on. This can happen, e.g., if no domain picks up a request with a high enough confidence threshold, or if there are no clarifying questions that could help to recover from the failure state.

## 3.3 Results on De-identified Historical Data

Table 2 shows the results of experiments on de-identified historical data, comparing relative-error and relative data-score efficiency metrics for VoG, random, and stratified sampling. Overall, the best

---

[15]This evaluation data consisted of roughly 3 million test cases that were sampled from user data and subsequently annotated by humans.

[16]In Çano and Bojar (2019), the relative data-score efficiency metric $\sigma$ was used to evaluate how well model accuracy scaled as the amount of training data was *increased*. We use use $\sigma$ in a slightly different but analogous manner to quantify how well model errors are avoided as the training data is pruned using a given sampling technique.

| Samp. technique | $\Delta$train | $\Delta$DCER $\downarrow$ | $\Delta$FSEMER $\downarrow$ | $\Delta$ICER $\downarrow$ | $\Delta$IRER $\downarrow$ | $\sigma_{\text{DCER}} \uparrow$ | $\sigma_{\text{FSEMER}} \uparrow$ | $\sigma_{\text{ICER}} \uparrow$ | $\sigma_{\text{IRER}} \uparrow$ |
|---|---|---|---|---|---|---|---|---|---|
| Random | $-52\%$ | 6.04% | 5.56% | 5.26% | 4.98% | $-11.7$ | $-10.78$ | $-10.19$ | $-9.64$ |
| Stratified | $-52\%$ | 5.23% | 5.14% | 4.87% | 4.92% | $-10.14$ | $-9.96$ | $-9.43$ | $-9.54$ |
| VoG-class-norm | $-52\%$ | 5.48% | 5.08% | 5.20% | 4.40% | $-10.53$ | $-9.76$ | $-10.01$ | $-8.47$ |
| VoG-dset-norm | $-52\%$ | **2.94%** | **3.65%** | **3.18%** | **3.34%** | **$-5.65$** | **$-7.02$** | **$-6.11$** | **$-6.42$** |
| VoG-class-norm | $-46\%$ | 3.62% | 3.77% | 3.63% | 3.39% | $-7.87$ | $-8.18$ | $-7.87$ | $-7.36$ |
| VoG-dset-norm | $-46\%$ | **1.52%** | **2.20%** | **1.72%** | **2.09%** | **$-3.25$** | **$-4.73$** | **$-3.69$** | **$-4.50$** |
| VoG-dset-norm | $-14\%$ | 0.53% | 1.02% | 0.53% | 0.97% | $-3.86$ | $-7.44$ | $-3.92$ | $-7.09$ |

Table 2: Comparison of offline-test error-rate and data-error efficiency metrics for models trained on in-house data, sampled according to different techniques. All pairwise comparisons were relative to a baseline trained on all data. For each technique, we report the average change in performance for three models trained on independently sampled training data.

performance for 52%-pruned data was obtained by models trained on VoG-*dataset*-norm-sampled data, while random sampling was associated with the worst model performance across all evaluation metrics. The stratified-sampling pruning baseline improved over random sampling, particularly with respect to domain-classification accuracy ($\Delta$DCER of 5.23% for stratified sampling vs. 6.04% for random sampling). In fact, except for $\Delta$FSEMER and $\Delta$IRER that track slotting errors, models trained on stratified-sampled data even slightly outperformed models trained on VoG-*class*-norm-sampled data.

The experimental results in Table 2 demonstrate the importance of score normalization: models trained on data pruned by dataset-normalized VoG scores outperformed models trained on data pruned by class-normalized VoG scores across all evaluation metrics we considered, for both pruning percentages. Using class-normalized scores as sampling weights increased overall relative DCER by roughly 1.9x when pruning 52% and by roughly 2.4x when pruning 46%, compared to when sampling from dataset-normalized scores. In App. C.5, we provide a detailed domain-level analysis to understand which data contributed most to the improvement associated with pruning by dataset-normalized VoG scores versus class-normalized.

Table 2 also shows that the efficiency metric $\sigma$ for dataset-normalized VoG-pruning was higher when pruning 46% compared to when pruning 52% or 14%. These findings can be used to help infer appropriate pruning targets for a given training dataset that minimize the need for historical training data without regressing on model performance.

### 3.4 User Study Results

Table 3 shows the results of our user study comparing the baseline model to a model trained on

| Metric | Rel. change | 95% CI |
|---|---|---|
| UER | $-1.18\%$ | $[-2.00\%, -0.35\%]$ |
| PDR * | $0.45\%$ | $[-0.18\%, 1.09\%]$ |
| PDR-tail | $0.56\%$ | $[0.003\%, 1.09\%]$ |

Table 3: Comparison of top-level UER, PDR, and PDR-tail metrics for baseline model vs. VoG model candidate that has been trained on roughly half of the live data used for training the baseline model.
* Not a stat. sig. difference ($p = 0.16$).

roughly 50% of historical data.[17]

The reduced-train-data model surprisingly achieved slight statistically significant *improvement* in overall UER and with no statistically significant change to PDR. We saw a small but statistically significant degradation in PDR-tail, which indicates that this type of aggressive data reduction can lead to regressions for less common requests. We also present a domain-level analysis of these top-line results in App. C.5.

Taken together, these results suggest that our NLU training data is highly redundant, and that comparable performance can be had by training on an intelligently chosen subset of it. While regressions in per-domain UER and PDR suggest potential downsides of pruning data based solely on DC-model gradients for all statistical models of a hierarchical NLU system, these results nevertheless confirm the overall practical utility of pruning data by VoG scores in commercial settings.

---

[17]As discussed in Section 3.1, the candidate model also included other non-live training data (e.g., synthetic) in model training, which was also the case for the baseline.

## 4   Discussion

In this work, we initiated a study of the application of influence scores in efficient language data sampling. First, we benchmarked a diverse subset of influence scores in a data reduction task and found promising results pruning using VoG scores. In the second part, we used these preliminary results on SNLI as impetus to scale VoG up to a model stack commonly used in commercial voice assistants. We provided a detailed account of how score normalization affects final results and again found encouraging results on experiments involving historical data using dataset-normalized VoG scores, as well as in a user study. In particular, we did not see any overall regressions when a model trained only on ~50% of historical data was deployed.

This work mainly focused on data reduction; it would be interesting to reverse some of the presented analysis for data mixing/augmentation in order to identify economical ways of surfacing new model data. Our work also focused on supervised settings using BERT architectures; an obvious path forward would be to extend the definition of these scores to model pretraining and to, e.g., decoder-only architectures that are commonly used for large language models (see, e.g., Marion et al. (2023) along similar lines). While this may be difficult to implement at a microscopic level for a corpus of pretraining data such as the Common Crawl, one avenue could be to apply this method at a coarse-grained level by grouping together texts by similarity. Given the recent results of Sorscher et al. (2022) and Abbas et al. (2023), this suggests a path towards training data-efficient large language models that could, in principle, outperform empirically observed scaling laws (Kaplan et al., 2020; Hoffmann et al., 2022).

Another highly promising application of our results is determining the influence of specific examples in in-context learning. One concrete generalization of VoG scores to this setting would be to look at variance of model weights (e.g., in attention heads) in specific layers over the length of the input sequence. This could provide an interpretable metric for identifying influential contextual examples and failure modes, at the level of specific tokens (Grosse et al. (2023) propose similar methods using influence *functions*). Given the increased recent interest in this area of research due to concerns over bias, toxicity, and fairness in large language models, there is a critical need for simple, inexpensive, and empirical metrics that can estimate the the influence of examples to in-context learning. Our work develops the foundational understanding necessary to make progress on that problem by generalizing results from the computer vision field (such as those scores that approximate more computationally expensive influence functions) to language-based tasks.

## Limitations

We hope that our in-house experiments provide a useful data point on the practical utility of influence scores. However, we note that we could not experiment with the same number of sampling techniques or prune sizes as we did in SNLI experiments due to computational overheads, and acknowledge that our in-house results are not readily reproducible. In addition, the customer data available for model training and experimentation changes frequently, e.g. due to data-expiration policies or customer data-deletion requests, which limited our ability to strictly control the training data between all related model-training runs. However, this limitation applied equally to each experimental sampling technique and only impacted the relative training-data reductions for a given pruning fraction by less than 0.01% for all sampling techniques.

We also note that the goal of our paper was not to find the absolute, best-performing influence score through extensive score-hyperparameter tuning. It is highly possible, for example, that the benchmarking measurements reported in Fig. 1 can be refined for better accuracy (though we have aimed to provide a thorough documentation in App. B.1 of our initial pass at score-hyperparameter tuning on SNLI). Instead, our results should be taken as a proof-of-concept of the existence – and applicability – of a simple, scalable, one-shot influence score in both public and production language data reduction settings.

Finally, our public data experiments primarily focused on a controlled setting using the SNLI dataset, which may not generalize to other public datasets. To address this, we conducted the scaled user study which exposed the model to unconstrained human speech, which varies (often dramatically) in carrier phrase frequency, vocabulary, named entities distribution and other aspects from publicly available datasets such as SNLI.

## Ethics Statement

Influence-based filtering can have disparate impact on predictions for classes with less annotated data. This can increase the likelihood of training data associated with less frequent language patterns being filtered out, which can increase bias in data that then propagates to the trained model. We have attempted to quantify this bias and correct it through, e.g., dataset-normalization.

## Acknowledgements

We are grateful for the helpful discussions and feedback provided by Jason Crowley and Kay Rottmann.

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

## A  Review of Influence Scores and Related Work

In this work, we use "influence scoring" as a broad term to refer to the large body of scientific literature focused on using artifacts of the learning algorithm – such as the loss, model confidence, etc. – to determine the relative importance of specific data instances. Many of these methods can be used to determine influential test examples, in addition to training. This review section should not taken to be comprehensive or exhaustive, but rather as a starting point to delve into subtopics in this area of research. We suggest the "Related Works" section in Swayamdipta et al. (2020) for a nice review of these methods as well.

There are a number of works that aim to use empirically formulated scores to approximate or improve upon influence *functions* – formulas that estimate the impact of training examples on test examples (see, e.g., Koh and Liang (2017) and references therein). TracIn (Garima et al., 2020) is one such example. Similarly, there are a number of methods that center around explainability

and interpretability; e.g., finding representer points by decomposing pre-activation predictions (Yeh et al., 2018), methods that aim to extract feature importance (Sundararajan et al., 2017; Lundberg and Lee, 2017), develop reliable models of predictions (Ribeiro et al., 2016), and capture learning order in neural networks (Hacohen et al., 2020).

Next, there is a body of literature that broadly includes methods that aim to quantify data quality and difficulty. This includes core-set methods that select an intelligently weighted subset of training data (Hwang et al., 2020; Har-Peled and Kushal, 2007), information-theoretic measures of data quality (Song et al., 2012; Ethayarajh et al., 2022), training dynamics based methods to diagnose and map out datasets (Swayamdipta et al., 2020; Agarwal and Hooker, 2022; Siddiqui et al., 2022), and papers that provide empirical and theoretical definitions of dataset difficulty (Meding et al., 2022; Baldock et al., 2021; Sorscher et al., 2022). Similarly, previous works have used adversarial filters (Bras et al., 2020) and proxy selection methods (Coleman et al., 2020) to score examples. A different approach taken by several works is to identify certain *types* of examples such as prototypical examples that match human expectations (Kim et al., 2014; Bien and Tibshirani, 2011), memorized examples (Liu et al., 2021), or outliers and tails in distributions (Carlini et al., 2019).

There are also several methods that leverage training dynamics to explicitly maintain/improve accuracy and learning efficiency (Hara et al., 2019; Pleiss et al., 2020b; Toneva et al., 2019; Paul et al., 2021; Jiang et al., 2019; Fayyaz et al., 2022; Jiang et al., 2020), and those that quantify bias in compressed models (Hooker et al., 2021).

In this paper, we could not exhaustively cover each of these scores, but as outlined in Sec. 2.1, we aimed to select a sufficiently diverse subset that could plausibly scale to a production stack.

## B  Additional Details about SNLI Experiments

### B.1  Score Implementations

Our implementation for the scores we tested in Table 1 aims to mirror the implementations given in the original references as closely as possible. Our experiments were mainly carried out on BERT$_{\text{SMALL}}$ ($L = 4$, $H = 512$, 29.1M parameters), trained for 10 epochs (with a batch size of 128) using the Adam optimizer with a learning

rate of $1 \times 10^{-4}$ for the encoder and $1 \times 10^{-3}$ for the classifier head. The classifier head was a three-layer fully connected neural network with an intermediate dimension of $64 \times 64$, with 10% probability drop-out. We specify our implementations below for the influence scores:

**VoG**: VoG scores were computed adhering closely to the method described in Agarwal and Hooker (2022), with the exception that the input pixels were replaced with the input embeddings. Gradients were computed at the locations of the ground-truth labels. The pseudo-code given in algorithm 1 describes our implementation in full. The computation is split into two steps for clarity: in the first, we compute the gradients of the pre-softmax model outputs at the location of ground truth labels with respect to the outputs of the embedding layer, for the desired number of model checkpoints $N_c$ (we used 10). For each example $i$, these gradients will be of dimension (input_length, embedding_dim), which we denote by $G_{ijk}^{(c)}$ where $c$ is the checkpoint, or in matrix form $\boldsymbol{G}_i^{(c)}$:

$$\boldsymbol{G}_i^{(c)} = \frac{\partial A_i^{(c)}}{\partial \boldsymbol{E}_i^{(c)}}, \qquad (2)$$

where $A_i^{(c)}$ denotes the pre-softmax model outputs at the location of the ground truth label and $\boldsymbol{E}_i^{(c)}$ denotes the embeddings. Next, the VoG score for each example $i$ can be computed by first computing the gradient means and variances across checkpoints:

$$\boldsymbol{\mu}_i = \frac{1}{N_c} \sum_{\text{checkpoints } c} \boldsymbol{G}_i^{(c)},$$
$$\boldsymbol{V}_i = \frac{1}{\sqrt{N_c}} (\boldsymbol{G}_i^{(c)} - \boldsymbol{\mu}_i)^2. \qquad (3)$$

The (unnormalized) score $v_i$ for each example is then given by the mean of $\boldsymbol{V}_i$ (that is, we average over the input embeddings, analogous to how the scores were averaged over pixels in Agarwal and Hooker (2022)). The final scores VoG$_i$ can be computed by normalizing $v_i$ with respect to either the score mean and standard deviation in each class, as originally prescribed in Agarwal and Hooker (2022), or the score mean and standard deviation for the full dataset:

$$\text{VoG}_i = \frac{v_i - \mu_{\text{class}}}{\sigma_{\text{class}}} \quad \text{(class-norm)},$$
$$\text{VoG}_i = \frac{v_i - \mu_{\text{dset}}}{\sigma_{\text{dset}}} \quad \text{(dataset-norm)}. \qquad (4)$$

VoG scores were computed in a "one-shot" manner, using gradients logged from a single training run.

---

**Algorithm 1** VoG implementation for language data.

---

load model $m$ used for training
gradients $G \leftarrow$ empty dict
**for** checkpoint $c$ in training checkpoints **do**
    load $m \leftarrow c$
    set $m$ to inference
    **for** batch $\in$ DataLoader **do**
        $x, y \leftarrow$ batch
        outputs $\leftarrow m(x)$
        get embedding layer $E$ from $m$
        set embeddings to retain grad
        $Y \leftarrow$ one-hot vector encoding of $y$
        compute back. pass on outputs w.r.t $Y$
        $G[c][\text{batch}] \leftarrow$ detached gradients of $E$
        zero-out model gradients
    **end for**
**end for**
VoG scores $v \leftarrow$ empty dict
**for** batch $b \in G[\cdot]$ **do**
    **for** example $i \in$ batch **do**
        $V \leftarrow \text{Var}(G[c][i])$ across checkpoints $c$
        $v[i] \leftarrow$ mean of $V$
    **end for**
**end for**
return $v$

---

**TracIn**: TracIn scores were computed using eq. 1 given in Garima et al. (2020), reproduced here for convenience:

$$\text{TracIn}(z, z') =$$
$$\sum_{i=1}^{k} \eta_i \nabla_w \mathcal{L}(w_{t_i}, z) \cdot \nabla_w \mathcal{L}(w_{t_i}, z'). \quad (5)$$

The content of the above equation is that TracIn computes a score for *pairs* of examples $z, z'$, such that high (low) scores correspond to proponents (opponents) to $z$. $\eta_i$ denotes the learning rates for checkpoints $i \in 1, \ldots, k$. Gradients are taken with respect model weights at these checkpoints. In our fine-tuning experiments, the learning rates are constant across checkpoints $\eta_i = \eta$, and thus enter as an overall factor to the scores that can be normalized away.

Following the original paper, for generating scores for training examples, we compute the *self*-influence scores i.e. we set $z' = z$. It is not tractable to compute the gradients with respect to

all of the model weights. A key question, then, is which layer the above gradients should be taken with respect to. We experimented with two possibilities: using the last-layer classifier weights and using the last encoder hidden layer. We found that in both clean and noisy SNLI settings, using the encoder hidden state gave more stable and better results for data pruning, relative to the random sampling baseline (this is what was used in Figs. 1 and 3). The final scores were computed from the L2 norm of the gradient dot products in eq. 5. Scores were computed using 10 model checkpoints from a single training run, logged every 500 iterations during training.

**Forgetting Scores**: Forgetting scores measure the number of times an example moves from being classified correctly to classified incorrectly. A key hyperparameter we had to tune was the cadence at which forgetting events are computed for each training example. In Toneva et al. (2019), this measurement was done at the batch-level granularity – that is, forgetting scores were updated each time the example was seen in the minibatch. We found that due to the rapid convergence of fine-tuning, this resulted in too many examples having a zero forgetting score. Fig 6 shows the distribution of forgetting scores for two different cadences; we see that the number of zero forgetting events increased by approximately 26% when the scores were computed every 500 iterations as opposed to every 50 iterations. To get enough resolution, we compute forgetting scores for the entire training dataset every 50 training iterations for the first 2 epochs of training, averaged over three random seeds in order to obtain sufficient precision in the head of the score distribution. For future work, it may be best to go to even finer resolution and compute forgetting scores *as frequently as possible* early in training, e.g., the first dozen iterations in training.

**EL2N**: EL2N scores were computed in the manner described in Paul et al. (2021) by the equation:

$$\text{EL2N}(z) = \|\text{softmax}[f(z)] - \boldsymbol{y}\|_2, \quad (6)$$

where $\text{softmax}[f(z)]$ indicates the softmax of the model outputs and $\boldsymbol{y}$ indicates the one-hot encodings of the labels.

Final EL2N scores were obtained by averaging scores over 10 training runs. Consistent with the findings of Fayyaz et al. (2022), we found that it is critical that the scores are computed early in

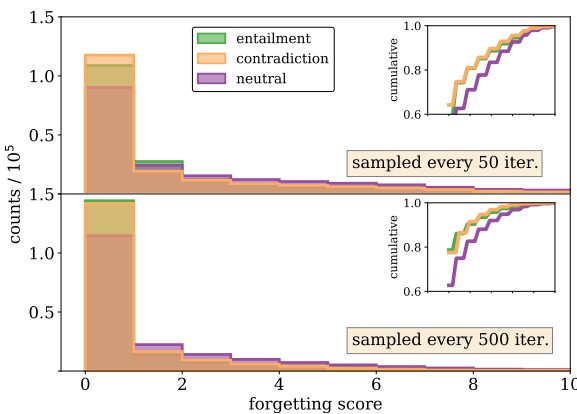

Figure 6: Distribution of forgetting scores for SNLI training data at two different sampling frequencies.

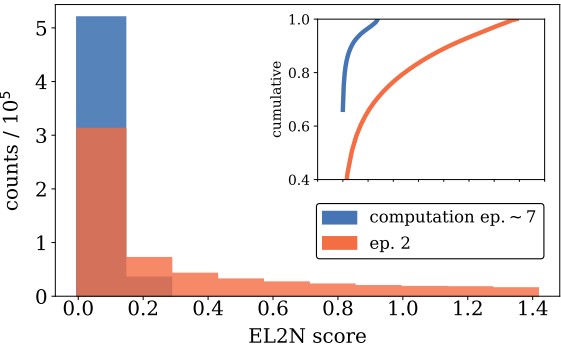

Figure 7: Distribution of EL2N scores for SNLI dataset at two different checkpoints.

training. We hypothesize that this is due to the rapid convergence of BERT on the training set; after only 6 epochs of training BERT$_{\text{SMALL}}$ has nearly memorized the training set (achieving close to 97% accuracy), which results in an EL2N score close to zero for many examples for which the margin is large (i.e. these examples are always learned). This is confirmed in the distribution of EL2N scores seen in Fig. 7, where there is a greater spread in EL2N scores computed at epoch 2 compared to epoch $\sim$7. When computed at epoch $\sim$7, the standard deviation of the scores corresponding to 50% of the head of the distribution (i.e. the easiest examples) is on the order of $10^{-4}$. $10^{-2}$ is roughly the mean standard deviation for individual scores *between* 10 re-runs, so the precision with which we can measure the score of an individual example is roughly on the order of $\sim 1/\sqrt{10} \times 10^{-2} \gg 10^{-4}$. This back-of-the-envelope calculation means that in order resolve the easiest examples correctly for scores computed *late* in training, one would need to average EL2N scores over an increased number of training re-runs.

**PVI**: Pointwise $\mathcal{V}$-information (PVI) scores from Ethayarajh et al. (2022) were computed using eq. 4 of their paper, reproduced here:

$$\text{PVI}(x \rightarrow y) = -\log_2 f[\emptyset](y) + \log_2 f[x](y). \tag{7}$$

The scores require fine-tuning a "null" model, denoted by $f[\emptyset](y)$ that is trained on empty or null inputs. $f[x](y)$ denotes the model fine-tuned on training data. Both models were trained for 2 epochs (we find that empirically this is approximately when the $\mathcal{V}$-information is maximized[18])

---

[18]Interestingly, we also find that at late training times, PVI

and final scores were obtained from averaging over 10 random seeds. The scores were computed for models trained on all of the data (and subsequent models were trained on pruned data according to those scores). In future work, it would be interesting to consider iterative pruning, where scores are recomputed for models trained on data pruned using the previous models' scores.

### B.2 Probabilistic Sampling vs. Hard Cut-Off in SNLI

Each influence score provides a ranking of examples that orders their importance. We considered two different strategies for selecting data once the scores are computed: **hard cut-off** and **probabilistic sampling**. For the hard cut-off method, we only retain examples with scores above a certain threshold (e.g., to prune 30% of the "easy" data, we would prune the 30% of examples corresponding to the head of the score distribution). The probabilistic method relaxes this condition, and each example has a chance of being retained with a probability equal to the softmax of its score. We used the probabilistic sampling method in two cases: first, in sampling from forgetting scores since this was a discrete score with a vast majority of examples sharing an example score of 0. Therefore, setting a hard cut-off would have removed all of these examples. Second, we used probabilistic sampling for dataset-normalized VoG scores, since pruning from the tail with a hard cut-off resulted in too many examples from the "entailment" class being removed (see Fig. 2). For our in-house experiments on customer data, we opted for linear probabilistic

---

and EL2N scores become correlated. This offers another explanation for why EL2N scores have be to computed early in training – the amount of *usable* bits of information decreases over the course of model training.

sampling instead of softmax sampling (described in Sec. C.3).

## B.3 Impact of Model Size

We investigated the effect of model capacity on pruning by VoG-sampling. Fig. 8 shows test accuracy versus percent of SNLI training data pruned, for both (class-normalized) VoG-score sampling and random sampling, for BERT$_{SMALL}$ and BERT$_{BASE}$ ($L = 12, H = 768$, 110.1M parameters) (Turc et al., 2019). The BERT$_{BASE}$ encoder was trained using the Adam optimizer with a learning rate of $0.9 \times 10^{-4}$, along with a 3-layer classifier head with an intermediate layer dimension of 256 and a learning rate of $0.95 \times 10^{-3}$, with 10% drop-out probability.

Aside from having a somewhat larger spread in final test accuracy, we see that the rough qualitative effect of the larger architecture is an overall shift in accuracy for each of the sampling methods. At 45% of the data pruned, sampling by VoG-easy on BERT$_{BASE}$ has a test accuracy of 86.70±0.8%, compared to BERT$_{SMALL}$ which had 85.04±0.2%. This provides some encouraging evidence that VoG-based pruning is useful for performance-efficient sampling of training data across different BERT models.

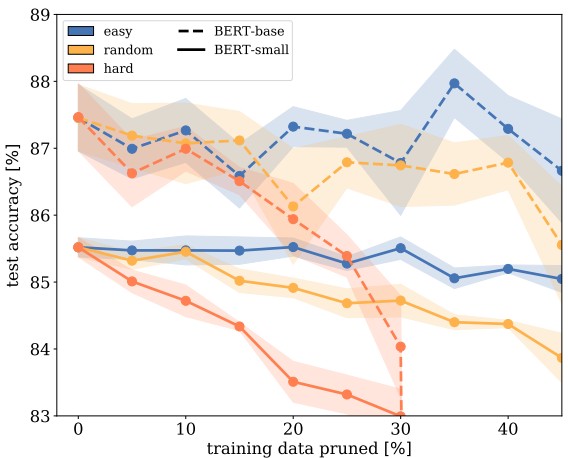

Figure 8: Shows test accuracy versus percent of SNLI training data pruned by VoG and random sampling, for both BERT$_{SMALL}$ and BERT$_{BASE}$. Each data point represents the mean over three training runs while shaded region shows the 1-$\sigma$ envelope.

## B.4 Encoder Representations of Scored Data

In our data pruning plots (Fig. 1), we observed a drop in test accuracy when pruning hard examples for most of the influence scores. In Sorscher et al. (2022), it is hypothesized that this happens because these examples are support vectors critical in forming the decision boundary between classes and removing them does not result in usable representations of the test data. Phrased differently, we have seen that most training examples can be removed without dramatically impacting test accuracy; the converse of this statement is that a small number of training examples have an outsized impact on test accuracy. We can visualize this explanation in the case of EL2N scores, which are explicitly defined to be the marginal distance between the model predictions and the one-hot encoded labels.

In Fig. 9, the left subplot shows the t-SNE representation of the SNLI training data, with five percent of the most difficult EL2N examples highlighted in red. The bulk of these difficult examples lies on the decision boundary between entailment and neutral classes. When none of the data is pruned, the center plot shows the t-SNE representation of the SNLI test data, comprised of three well-defined clusters. When the most difficult EL2N examples are removed from the training set, we see that the representation of the test data (rightmost subplot) is comprised of a less defined clusters of roughly uniform density. In particular, the boundary between contradiction and neutral classes almost completely disappears, indicating that the model cannot resolve the differences between the two classes as well as in the no-pruning scenario.

## B.5 Score Distributions and Examples in SNLI

Score distributions for SNLI are shown in Fig. 10. Examples from the head and tail of each of these distributions is given in Tables 11 through 15.

## B.6 Noisy Data-Reduction Experiment Details

For experiments on SNLI with added noise, we chose to experiment using VoG, TracIn, and PVI scores. These were selected because VoG outperformed other metrics in the non-noisy data reduction setting, and TracIn and PVI due to their potential efficacy in identifying misannotated data.

Noisy versions of the SNLI datasets were created by shuffling of labels in an isotropic manner, which meant there was a chance (roughly 30%) that any given label would not flip. Therefore, the label noise quoted in Fig. 3 should be taken as an upper

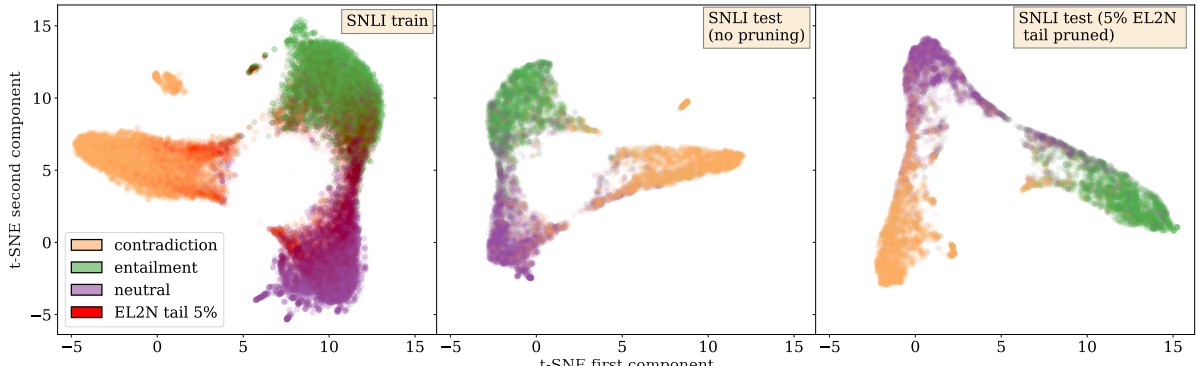

Figure 9: t-SNE plots of EL2N for the training data with 5% of the EL2N highlighted in red (left), test data before pruning the tail 5% of EL2N examples (center), and test data after pruning.

bound to the true number of misannotations[19], as quantified in Table 4.

| % noise added | # misannotations | % of train data |
|---|---|---|
| 5 | 18433 | 3.35 |
| 10 | 36693 | 6.67 |
| 15 | 55083 | 10.01 |
| 20 | 73724 | 13.40 |
| 25 | 91535 | 16.63 |
| 30 | 110189 | 20.02 |

Table 4: Shows the number of misannotations for each version of noisy SNLI data, determined by comparing against the version of SNLI with no added label noise. The last column shows the percent of mislabeled training data.

## C Additional Details about In-House Experiments

### C.1 Overview of Common Commercial NLU Systems

The natural language understanding (NLU) component of common commercial systems consist of deterministic systems that cover the most critical and frequently occurring utterances (e.g., "stop!"), while other utterances are covered by multiple statistical ("stat") models organized in a hierarchical fashion. For example, an utterance spoken to a conversational assistant not covered by deterministic artifacts will typically first be classified to an appropriate domain by a BERT-based domain classifier (DC) model, then a specific intent within that domain by intent classifiers (IC) models, followed

---

[19]In historical data, we often do not have an exact count of the misannotations, but only a rough estimate of the overall noise.

by named entity recognition (NER) to resolve entities (such as city names, times, etc.) within the utterance. Each BERT-based statistical model was trained on spoken-form Japanese data. Our experiments focus on the fine-tuning stage of model training, performed on in-house data.

In our experiments we computed VoG scores for the DC model, but we measure the performance impact of doing so for the composite hierarchical model, including the impact on the accuracy of downstream IC and NER models.

### C.2 Distribution of Types of Training Data

Model training data is collated from multiple, varied sources (e.g., synthetic, human-annotated, weak-signal). Data from different sources may have different label-noise distributions and class distributions, which in turn can impact influence scores computed on that data. Table 5 shows the distribution historical and synthetic data in in-house data.

| Domain | % of total | % historical | % synth. |
|---|---|---|---|
| Music | 18% | 89% | 11% |
| Home Automation | 12% | 89% | 11% |
| Knowledge | 11% | 85% | 15% |
| Notifications | 7% | 86% | 14% |
| Video | 6% | 74% | 25% |
| Shopping | 5% | 93% | 7% |
| Health & Fitness | <1% | 4% | 96% |
| Overall | 100% | 81% | 19% |

Table 5: Distribution of data sources used for training the model used for in-house experiments.

In in-house experiments, training-data pruning was performed on de-identified historical user data. In the experiments on historical data, this was the only training data used; models were fine-tuned

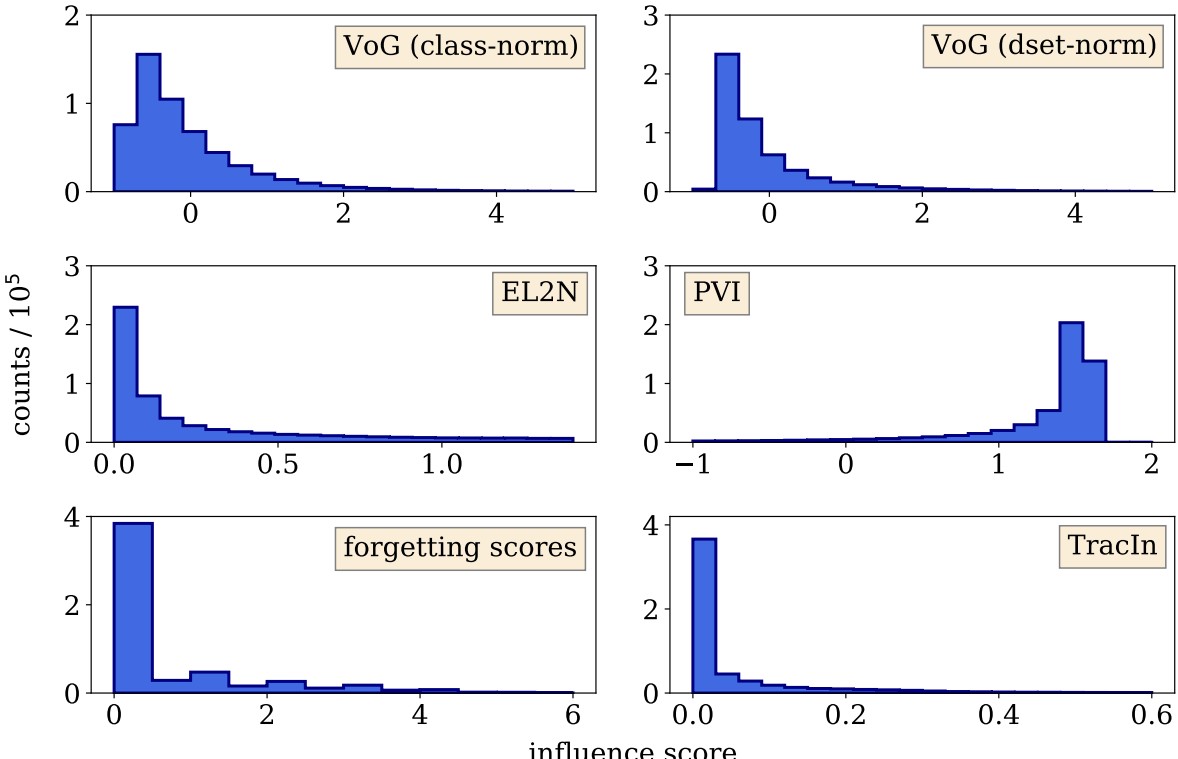

Figure 10: Shows the distribution of influence scores in Table. 1 computed for SNLI data.

solely on user data. In the user study experiments, stat-model training data was performed on historical data as well as additional supplemental data (e.g., resampled, synthetic, etc.).

This difference was motivated by practical concerns. For some smaller (e.g. newly introduced, or low traffic) NLU domains, the amount of historical data available for model training was small in size (less than 20 instances). This combined with domain-wise data imbalance led to regression in that subset of domain. That was fine in offline analyses (since it applied to all pruning conditions) but unacceptable for exposure to real users.

Due to the cost and operational overheads involved with running the user study, we could not try the same number of sampling techniques as we did for the historical data experiments. In our user study, we tested our most promising technique from the historical data experiments, sampling by dataset-normalized VoG scores, and compared relative to the baseline model.

### C.3 Filtering Train Data via Score-Weighted Sampling

Our approach for filtering in-house data based on influence scores used a slightly different approach than the softmax probabilistic sampling described

in Appendix B.2.

The motivation behind this was that we did not have a robust characterization of the noise in our customer data and found that softmax sampling was too aggressive in downsampling easy-to-learn utterances (and perhaps retaining too many noisy, hard-to-learn examples). In order to preserve a larger fraction of these easiest utterances, a sampling approach was used where the probability of sampling was *directly proportional* to the VoG score. This was accomplished by linearly transforming normalized VoG scores $\text{VoG}_{norm}$ (including negative and positive values) to the range $[\epsilon, 1]$. The positive-transformed scores were them normalized by dividing by the sum of all positive-transformed scores in order to produce sampling probabilities. This type of transformation aims to preserve the relative ratios between old and new values that existed pre-transformation. That is, if the VoG score of Utterance A was twice the VoG score of Utterance B, the sampling probability for Utterance A will be approximately twice the sampling probability for Utterance B.

Finally, in order to filter training data by sampling scores we first decide on a proportion of data to prune. This in turns determines the number of training examples to sample via weighted random

sampling without replacement. See Figure 12 for an example of 50% train-data reduction using this method.

## C.4 VoG Distributions on In-House Data: Additional details

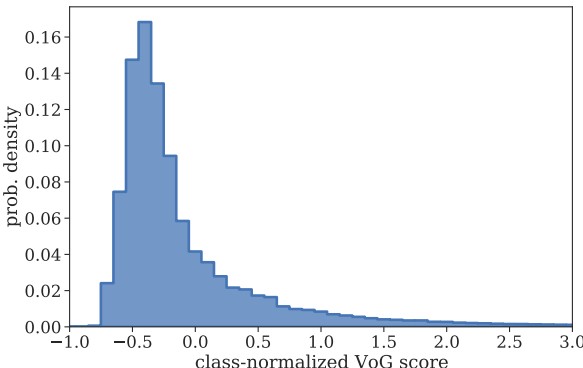

Figure 11: VoG scores for 10.9 million de-identified historical data instances downsampled to train candidate models.

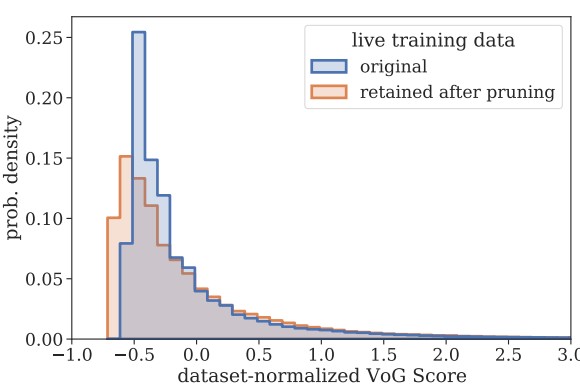

Figure 12: Dataset-normalized VoG scores in the original training data vs. for the -52%-reduced subset obtained via VoG-score-based sampling. VoG scores for the reduced-size subset have been re-normalized according to the mean and std of the reduced-size subset scores.

Fig. 11 shows the distribution of VoG scores for the subset of historical training data used to train the statistical models, which comprises a majority of the overall training data. Compared to the SNLI VoG distribution in Fig. 2, the VoG distribution of the historical data looks roughly similar, but has more power in the low-scoring, "easy" bins. While in historical data 72% of class-normalized scores were less than 0, for SNLI only 66% of scores were less than 0.

Fig. 12 shows the distribution of data retained using dataset-normalized VoG scores. Figure 13 shows the VoG distribution using dataset-

normalized VoG scores for the same subset of domains shown in Figure 4.

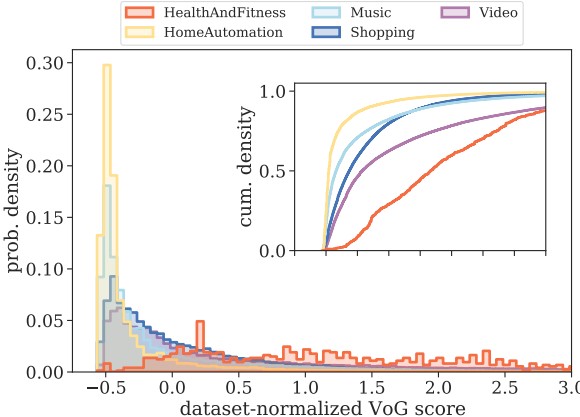

Figure 13: Dataset-normalized VoG scores for historical data downsampled to train candidate models, grouped by ground-truth domain. *(Top)* Cumulative probability-density distribution. *(Bottom)* Probability-density distribution for the same data, using a score bin size of 0.05.

VoG scores appear to capture more than just the influence of a domain's data related to the size that domain's data, but also align with the intuition that training data associated with domains that have more complex domain definitions provide more of a challenge for the model to predict correctly, and that training on these challenging examples is more likely to influence the learned model parameters than training on easy examples would.

For example, while HomeAutomation and HealthAndFitness training data are similar in intent-label diversity (2.3 bits of entropy for both), they differ greatly in training-data representation (12% vs. <1%). Intuitively, we would expect smaller domains such as HealthAndFitness to exhibit higher average VoG scores than larger domains such as HomeAutomation, which we indeed find. The median class-normalized VoG score for HomeAutomation was -0.31 compared to a median of -0.22 for HealthAndFitness). As shown in the top of Fig. 4, a larger proportion of HomeAutomation training instances were associated with negative VoG scores than for HealthAndFitness (roughly 80% vs. 60%).

While Shopping and Video constitute similar proportions of the training data (6% vs 5%), the intent-label distribution for Shopping exhibits greater diversity than the intent-label distribution for Video (2.5 vs. 1.6 bits), indicating a more complex and difficult prediction task. This difference in intent diversity appears to be reflected in class-normalized VoG scores; Shopping scores tend to

be higher (more positive) than Video scores (median of -0.30 in Shopping vs. -0.39 in Video). As shown in Fig. 4, VoG scores for Video data were more densely located in the low-scoring, "easy" region; 37% of Video vs. 32% of Shopping data were associated with VoG scores below -0.5.

In some cases, it can be difficult to reason about the relative influence of one domain's data on a trained model compared to another domain's data. For example, in our real-world setting, the Music and Video domains capture similar NLU functionality (media playback) and are associated with comparable intent-label entropy estimates (1.5 vs. 1.6 bits). Unlike for HealthAndFitness, both Music and Video out-represent the majority of other domains in the training data (Music is the second largest domain at 18%, while Video is the eighth largest at 6%). In this case, which domain provides redundant or extraneous data not needed in order to maintain model performance? Appeals to intuition may fail us here, but VoG scores can still be helpful. As shown in Fig. 4, we find that Video training data contains a larger proportion of low-influence data instances than Music (37% of Video vs. 2% of Music of VoG scores were less than -0.5), potentially signaling the existence of redundant or duplicate Video training instances.

VoG-score summary statistics for de-identified historical training data used to train the candidate model are shown in Table 6 (normalized by class) and in Table 7 (by dataset).

| Domain | Median | Mean | Std |
|---|---|---|---|
| Health & Fitness | -0.22 | 0.0 | 1.0 |
| Notifications | -0.27 | 0.0 | 1.0 |
| Knowledge | -0.28 | 0.0 | 1.0 |
| Shopping | -0.30 | 0.0 | 1.0 |
| Home Automation | -0.31 | 0.0 | 1.0 |
| Music | -0.35 | 0.0 | 1.0 |
| Video | -0.39 | 0.0 | 1.0 |

Table 6: Class-norm VoG scores by domain, for a subset of domains in in-house training data. Within each class (domain), the mean score is 0 and the standard deviation is 1.

## C.5 Domain-Level Analysis of In-House Experiments

Per-domain offline results from the user study experiment are shown in Table 8. Per-domain user study results are shown in Table 9.

| Domain | Median | Mean | Std |
|---|---|---|---|
| Health & Fitness | 1.17 | 1.52 | 1.63 |
| Video | 0.08 | 0.86 | 2.00 |
| Knowledge | -0.12 | 0.11 | 0.83 |
| Shopping | -0.13 | 0.15 | 0.93 |
| Music | -0.34 | 0.10 | 1.27 |
| Notifications | -0.44 | -0.29 | 0.54 |
| Home Automation | -0.45 | -0.24 | 0.69 |

Table 7: Dataset-normalized VoG scores by domain, for a subset of domains in in-house training data. Across all domains, the mean score is 0 and the standard deviation is 1.

**Experiments on Historical Data**: In our experiments on de-identified historical data, increased representation of smaller domains when sampling by dataset-normalized scores translated to improved DC and IC recall. For HealthAndFitness, dataset-norm VoG sampling was associated with $\Delta$DCER of 4% vs. 29% for class-norm sampling. A primary contributor to improved DC and IC recall were improvements in Video, which under dataset-normalized sampling increased in training-data representation by relative 43% but under class-normalized sampling decreased in representation by relative 1%.

For domains such as HomeAutomation and Notifications that decreased in representation when sampling by dataset-norm scores, models trained on dataset-normalized VoG scores were associated with improved DC performance and comparable downstream NLU performance compared to class-norm models where those domain's training-data representation actually increased.

**User Study**: We analyzed the per-domain results to understand which domains/intents contributed to the observed top-level UER and PDR relative changes (Table 9).

A primary contributor to the observed top-level UER improvement were Video-related requests. Video UER decreased by relative -11.6%. Post-experiment deep dives show that for the VoG model requests that previously were classified as Video were now classified as Music or Knowledge. We saw that Music traffic slightly increased (rel. +1%) without an associated increase in Music UER, suggesting the majority of requests newly interpreted as Music in the VoG model could by served by the Music domain. On the other hand, Music PDR increased by relative 2.6%, which was a primary

| | Dataset-norm | | | | | Class-norm | | | | |
|---|---|---|---|---|---|---|---|---|---|---|
| Domain | Δtrain | ΔDCER | ΔFSEMER | ΔICER | ΔIRER | Δtrain | ΔDCER | ΔFSEMER | ΔICER | ΔIRER |
| Music | < 0% | -4% | 4% | -3% | 4% | 1% | -3% | 5% | -1% | 5% |
| Knowledge | 27% | 12% | 4% | 12% | 4% | <0% | 16% | 6% | 16% | 6% |
| Video | 43% | -1% | 2% | 1% | 1% | -1% | 9% | 4% | 7% | 4% |
| Shopping | 18% | -11% | 4% | -8% | 4% | -5% | -3% | 6% | -1% | 5% |
| Notifications | -37% | -2% | 2% | 2% | 2% | -7% | 22% | 2% | < 0% | 1% |
| HomeAutomation | -29% | -6% | 1% | -3% | 1% | 2% | -16% | 4% | -8% | 3% |
| HealthAndFitness | 84% | 4% | 3% | 6% | 5% | -6% | 29% | - | 35% | - |

Table 8: Comparison of per-domain offline-test relative-error-rate metrics for models trained on historical data, sampled according to class-normalized or dataset-normalized VoG scores. Results were averaged over three training runs.

| Domain | Δtraffic | ΔUER | ΔPDR |
|---|---|---|---|
| Music | 1.0% | -2.1% | 2.6% |
| Knowledge | 5.3% | 0.6% | 0.7% |
| Video | 9.0% | -11.6% | 1.9% |
| Shopping | -2.6% | 12.8% | 4.0% |
| Notifications | 1.2% | 6.6% | 3.0% |
| HomeAutomation | -1.1% | -5.0% | -2.9% |

Table 9: Relative change to UER, PDR, and domain traffic for a subset of domains in the the user study, comparing a baseline model to a version of that model trained on only 50% of historical training data. Model results reflect metrics for the full NLU system comprised of statistical and non-statisical model components. These relative-change results were not compared using hypothesis tests and may not be statistically significant.

contributor to the observed increase in top-level PDR. The simultaneous decrease in Music UER but increase in Music PDR suggests that Music domain could provide some kind of interpretation for a request, but that overall those interpretations were associated with a slight increase in defect rate (e.g. defects related to searching for the wrong artist name, or searching for an album rather than an artist).

The user-study impact on VoG-based pruning on the performance of individual domains was not solely determined by whether or not a given domain's representation was increased/decreased in training data. Both HomeAutomation and Notifications increased in their training-data representation (Figure 5), however we observed opposite-direction impacts in metrics for those domains. HomeAutomation was associated with reduced traffic (rel -1.1%), UER (rel. -5.0%), and PDR (-2.9%); whereas Notifications was associated with increased traffic (rel. 1.2%), UER (rel. 6.6%) and

PDR (rel. +3.0%).

## C.6 Label-Value-Trail and Intent-Label Entropy

In the experiment on in-house data, VoG scores were measured with respect to DC model gradients, which can overlook training utterances that are influential especially for training IC-NER models. In order surface training utterances influential for not only DC model but also IC-NER model training, we combined DC Vog scores with slot-label-trail entropy estimates, which provide a coarse-grained intent-level estimate of NER difficulty.

As an example, consider the customer request "play music by Howard Shore please", which has the annotation "Play|Other music|Other by|Other Howard|ArtistName Shore|ArtistName please|Other". We define the slot-label-trail as the sequence of slots labels and non-Other slot-label values for a given annotation. The slot-label-trail for this example would be "Other Other Other Howard|ArtistName Shore|ArtistName Other". The frequency of each such slot-label-trail found in the training data is used to compute Shannon entropy estimates for each intent, given by:

$$H_{\text{intent}}(T) = - \operatorname*{\mathbb{E}}_{t \sim P} \left[ \log_2 P(t) \right], \qquad (8)$$

where $P(t)$ is based on label-trail frequencies found in training data, and $T$ is all label trails in a given intent. Eq. 8 is the formula for computing the Shannon entropy of the distribution of annotation-label-value trails in the training data and is computed separately for each domain-intent combination. Since the log is computing using base 2, resulting entropy estimates are measured in units of bits.

The intent-label entropy can be calculated at a

domain level in a similar manner. These results are summarized in Table 10.

| Domain | Intent-label entropy (bits) | Slot-trail entropy (bits) |
|---|---|---|
| Music | 1.5 | 14.8 |
| Knowledge | 0.1 | 14.5 |
| Video | 1.6 | 13.8 |
| Shopping | 2.5 | 12.9 |
| Notifications | 2.1 | 10.2 |
| Home Automation | 2.3 | 8.5 |
| Health & Fitness | 2.3 | 6.1 |

Table 10: Domain-level intent-label entropy calculations and intent-level slot-label-trail, for a subset of domains in historical data. For intent-level entropy, the weighted average across intents is reported per domain.

We used the same approach to calculating sampling scores from entropy estimates as we did in the case of converting VoG scores to sampling scores. For a given utterance, the final sampling score was the average of the VoG-based and entropy-based sampling scores. weight. In offline tests, we found that this modification to sampling scores helped to slightly improve composite model performance as measured by SEMER (rel. -1.39%) and IRER (rel. -1.5%), which was reflected in per-domain improvements for the largest-traffic Music and Video intents. This came at the expense of a slight increase in Shopping slotting errors on Shopping offline tests (rel. +6.1%).

| Premise | Hypothesis | Gold Label | Score Measurement |
|---|---|---|---|
| **VoG (class-normalized) Tail (hardest)** | | | |
| An African woman is standing near a well filling a jug near a hut with a thatched roof. | A woman fetching water from a well beside an African Hut with a straw roof. | - | 1e4 |
| A man in black is applauding a runner wearing a red jersey and the number 281. | The man is running | - | 1e4 |
| People sitting down by a stream of water. | They are enjoying the stream outdoors. | - | 1e4 |
| the man is celebrating a big win and he opened a bottle of wine, but it squirted out all in the air. | A man is celebrating his sports team win with alcohol. | - | 1e4 |
| A skier is skiing on a snowy slope. | A person is enjoying gravity and snows low friction to increase velocity down an incline. | - | 1e4 |
| A bearded man in a black coat and black cap stands near a young girl who is inside a large brown open container. | The man does not like to shave. | - | 1e4 |
| A girl in a blue sweater sits on a person's shoulders and carries a pinwheel. | A girl enjoying with a topy on a person's shoulders | - | 1e4 |
| A girl in a silk shirt plays pool | A woman in a green skirt is in a bar. | - | 1e4 |
| A group of three people pose for a picture with smile while another sits aside without smiling. | Three guys are smiling, while a fourth is running away. | - | 1e4 |
| A man working on a ticket machine as two women stand near. | A man is giving out tickets. | - | 1e4 |
| **VoG (class-normalized) Head (easiest)** | | | |
| A man in a black coat and gray scarf walks on a sidewalk in between a fence and a row of hedges. | A woman eats her lunch in the cafeteria. | C | -0.880 |
| Two women are singing on a stage while a man in red plays the guitar. | The three people are riding jet skis in the ocean. | C | -0.880 |
| A construction crew in orange vests working near train tracks. | The crew is eating dinner at a seaside restaurant. | C | -0.880 |
| a woman wearing a black helmet riding on a bike | A man is sleeping alone in bed. | C | -0.880 |
| A black and white dog with a spotted face is running through a dirt field. | A dog is sleeping in his bed by the fireplace. | C | -0.880 |
| Different people are walking on a sidewalk, in both directions, in front of an orange canopy. | People are sitting on a couch watching tv. | C | -0.879 |
| Four dogs stand in the snow. | A cat is sleeping on the fridge. | C | -0.880 |
| A male is standing on a base pitching a ball. | The woman is singing in the shower. | C | -0.880 |
| A backpacker points to the snowcapped mountains as he stands on a rocky plain. | A woman is laying in her hospital bed. | C | -0.880 |
| A woman in a wetsuit in surfing in the ocean. | A woman is sleeping in her bed at home. | C | -0.879 |
| **VoG (dataset-normalized) Tail (hardest)** | | | |
| MMA fighter in white shirt with Black and white shorts practices a kick in the gym. | An MMA fighter is using a punching bag. | C | 24.921 |
| Glossy red apples lay stacked in a metal bowl. | There are a bunch of pears on a plate. | C | 26.670 |
| A woman and a girl are sitting on a tile floor behind a wooden rack for weaving. | The lady and her daughter are riding horseback. | C | 41.494 |
| An elderly white man is playing a trumpet into a microphone. | An elderly man plays yellow drums. | C | 25.417 |
| Two people with hard hats and orange vests are working. | The people with the hard hats on have yellow vests on as well. | C | 48.367 |
| Two women are drinking from yellow cups and laughing. | The two women in the corner are drinking from red cups and laughing. | C | 45.679 |
| A group of people, of all ages, listening to a concert being performed by a solo practitioner. | A single muscian entertains a diverse audience | E | 36.737 |
| a teenager is wearing a gray hooded top and some red beads around her neck. | The teenager is wearing a red beaded necklace. | E | 32.429 |
| The cat is squinting. | There are cats playing with yarn. | C | 28.069 |
| A boy on a skateboard grinding down a handrail. | A boy does a bike trick at the park. | C | 27.373 |
| **VoG (dataset-normalized) Head (easiest)** | | | |
| A boy in camouflage pants and his ball lying in front of a blue car. | A boy is laying on the sidewalk next to the car. | - | -0.714 |
| An asian child sits on the ground eating from a bowl outside a hut on a sunny day. | An asian child sits on the ground eating from a bowl is inside the hut | - | -0.714 |
| Two men are using two horse to help with farm work. | The men are at a farm | - | -0.714 |
| Three people skydiving, one of them is sticking there tongue out making a silly face. | The people skydiving are scared and shaking. | - | -0.714 |
| A girl in a blue sweater sits on a person's shoulders and carries a pinwheel. | A girl enjoying with a topy on a person's shoulders | - | -0.714 |
| Young boy with long blond-hair running with a large bouncy ball and had two people playing piggyback chasing him. | The boy is playing a game with the other two people. | - | -0.714 |
| Man is demonstrating to a student. | Man showing student how to paint | - | -0.714 |
| A man wearing a helmet is riding his bike down rocky terrain. | a man is chasing after a deer | - | -0.714 |
| Three young adults building a large sand castle at the beach. | The adults are by the beach. | - | -0.714 |
| A girl on monkey bars. | She has climbed up. | - | -0.714 |

Table 11: Shows the highest and lowest scoring VoG examples from SNLI. Gold labels are denoted by (C)ontradiction, (E)ntailment, (N)eutral, or - which indicates that annotators could not come to a consensus on the label. A numerical cutoff of $10^4$ was used to truncate high VoG scores.

| Premise | Hypothesis | Gold Label | Score Measurement |
|---|---|---|---|
| **EL2N Tail (hardest)** | | | |
| two teams playing rugby, there seems to be more of one team than the other in the picture. | Two teams are sitting inside watching a movie. | - | 1.412196 |
| A group of men haul black trash bags across the beach, nearby a pickup truck and a bulldozer. | Two women are swimming in the ocean. | N | 1.412198 |
| Children playing on a merry-go-round on a chilly day. | The children are sleeping | N | 1.412257 |
| A man is working on a laptop computer in an open air cafe. | A man is surfing the web on the couch. | N | 1.412444 |
| Two men are practicing a dance routine while their friend captures it on camera. | Two men are sleep | N | 1.412286 |
| A man with a red beard pushing a shopping cart on a busy street. | A dog sits inside a shopping cart. | N | 1.412446 |
| A person doing a jump with their snowboard over a orange and white caution sign. | A person is sleeping | N | 1.412697 |
| A girl in pink stands in front of a Mud coffee truck. | A group of boys playing marbles. | N | 1.412732 |
| Three ice skaters round a corner. | The skaters are sleeping. | N | 1.412452 |
| A male and female are at a table with a drink. | Two cats are at a table. | E | 1.412773 |
| **EL2N Head (easiest)** | | | |
| a brown and white dog jumping over a red, yellow and white pole | a cat sleeps on a pillow | C | 0.000646 |
| A dog with a ball that is running in a field. | Cats sleeping on a porch. | C | 0.000657 |
| A dog runs on a field with its mouth open. | Two cats sleep in a bed. | C | 0.000600 |
| A large brown dog is running through a grassy backyard. | A cat sleeps inside. | C | 0.000646 |
| A brown dog is crouching and looking up in a field of grass. | A cat is inside sleeping on a couch. | C | 0.000630 |
| Two dogs are fighting over a red Frisbee outside. | Two cats sleep on a shelf. | C | 0.000610 |
| A boy wearing a red sweater runs along a colorful beach. | A girl sleeps on the couch in front of a TV. | C | 0.000650 |
| Two dogs playing with a small blue ball in a grassy field. | two cats are sleeping inside. | C | 0.000663 |
| Two dogs run in a field of brown grass. | the cats are sitting on a couch | C | 0.000665 |
| A brown and white dog runs through a grassy area. | A cat sits in the living room. | C | 0.000668 |

Table 12: Shows the highest and lowest scoring EL2N examples from SNLI.

| Premise | Hypothesis | Gold Label | Score Measurement |
|---|---|---|---|
| **PVI Tail** | | | |
| A large crowd of people are outside and a big sign reads "AIDS WALK" in the background among trees. | The crowd marches in front of the sign. | - | 3.094221 |
| Girl hangs up in midair by two bungee cords. | A GIRL IS SWINGING FROM A ROPE. | - | 3.095381 |
| A blond woman tips a young blond girl upside-down. | The woman is holding the girl upside-down over the pol. | - | 3.170588 |
| A man is gathering hay on a horse drawn wagon. | Three horses are pulling a wagon covered in hay | - | 3.120966 |
| A rugby player wearing white short and shirt being tackled by another player wearing blue shorts and trunks; a teammate of he player being tackled is coming up behind the tackled player. | A rugby player wearing white short and shirt being tackled by another player wearing blue shorts and tree trunks | - | 3.123245 |
| The young boy sleds down the hill in the snow. | A boy is at the top of a snow-covered hill. | - | 3.183291 |
| A man in a white jacket standing in front of an older woman in a white jacket playing crochet. | The man was playing crochet with the two women. | - | 3.322716 |
| A crowd of people at a market near a highway. | There are 2 people in the marker | - | 3.187976 |
| A antiquated soldier stands in salute holding a rifle. | A statue of a soldier with a rifle. | - | 3.221569 |
| Five people are on their yard with two of them climbing a ladder to a tree in the background. | Two people are climbing a tree behind three other people. | - | 3.283761 |
| **PVI Head (potentially ambiguous or misannotated)** | | | |
| A group of people sitting outdoors. | The people are inside. | E | -11.263250 |
| Three dogs, two black one brown, are playing in a grass field. | Three cats, two black one brown, are playing in a grass field. | E | -13.630976 |
| Lacrosse players struggling for control of the ball. | Nobody is in control of the ball. | E | -12.107733 |
| A bunch of people sitting outside a building at night. | the dogs were fighting to get a bone from each other | E | -14.055565 |
| A male and female are at a table with a drink. | Two cats are at a table. | E | -12.030568 |
| A group of men in a blue car driving on the track. | One woman is driving the blue car. | E | -11.397809 |
| A man jumps into a bed that is set up on a public walkway. | A female leaps on a bed in a public walkway. | E | -11.016526 |
| People at a marketplace selling watermelons. | Dogs at a marketplace selling watermelons. | E | -10.657009 |
| Three people are using a net on a beach. | Three people are sitting on a bench. | E | -11.001061 |
| A little boy sits on the tail of a fake alligator. | The woman is sitting on a fake alligator. | E | -10.811694 |

Table 13: Shows the highest and lowest scoring PVI examples from SNLI.

| Premise | Hypothesis | Gold Label | Score Measurement |
|---|---|---|---|
| **Forget Scores Tail (hardest)** | | | |
| A man with a white shirt, brown shoes, and blue socks is reading near other people and piles of books. | The man is fully clothed. | E | 6.67 |
| The woman is sitting in the boat being rowed by the man. | The woman is sitting on the beach. | C | 6.67 |
| A lady is in a wheelchair on the corner of a street, and a man in a white shirt is about to cross the street. | There are people waiting to cross the street. | E | 6.67 |
| A woman carrying shoes is walking barefoot on the beach. | A woman walks barefoot by the ocean | E | 6.67 |
| Two men sit next to each other on the water in a boat. | The boat is floating. | E | 6.67 |
| Toddler boy wearing airplane shirt, in stroller with large beads. | The toddler is riding the stroller. | E | 7.0 |
| A woman with glasses is looking at a gray laptop with two other people at a table in a yellow room. | Three people are having lunch at an upscale restaurant. | C | 7.33 |
| Lady wearing a green work helmet and holding a bag. | The woman is wearing a straw hat. | C | 7.0 |
| Men and women in Scottish garb play the bagpipes as a runner approaches on the street. | People wear traditional clothes and play the bagpipes | E | 6.67 |
| This man is fit and well toned running enthusiast. | This man likes to run. | E | 7.67 |
| **Forget Scores Head (easiest)** | | | |
| A woman with black hair is typing on a laptop computer. | A woman in front of computer. | E | 0.0 |
| A woman wearing a blue shirt typing on a laptop. | Someone is typing. | E | 0.0 |
| Woman in black shirt walking along pier. | The woman is outside. | E | 0.0 |
| Woman in black shirt walking along pier. | The woman is walking to meet a friend. | N | 0.0 |
| A woman with black hair is typing on a laptop computer. | A woman typing in the computer. | N | 0.0 |
| A married woman in blue and black types. | A single mother on the run. | C | 0.0 |
| A woman with black hair and jewelry on her left hand and arm typing on a keyboard. | A woman has hair. | E | 0.0 |
| In a white-walled gymnasium, a shirtless male gymnast does a handstand on the parallel bars as another man watches him in the background. | In the school's gymnasium, a gymnast performs a handstand that scores a perfect score by the men rating him. | N | 0.0 |
| A married woman in blue and black types. | A woman works on her novel. | N | 0.0 |
| A woman wearing a blue shirt typing on a laptop. | Someone is replacing the battery on a laptop. | C | 0.0 |

Table 14: Shows the least and most forgotten examples from SNLI.

| Premise | Hypothesis | Gold Label | Score Measurement |
|---|---|---|---|
| | **TracIn Scores Tail (hardest)** | | |
| Three people playing rock music, one of them is wearing a hat and a black pants with a black t-shirt and singing and playing the guitar. | Out of three people playing music, one with black shirt and pants, playing piano. | E | 3.30 |
| A lady is giving a presentation to an all-female class. | There are no men in the room. | E | 3.30 |
| A man in a gray shirt is drilling into a silver can. | A lady drilling a can. | E | 3.32 |
| Several people on a stage with a blue background. | The humans are on the ground. | E | 3.40 |
| Three female bikers travel swiftly through a school zone. | two boys ride skateboard | E | 4.20 |
| A older man holds an object and is looking at two young girls that sit across from him at a table that is outside. | A man is holding an object, but is not looking at it. | E | 3.76 |
| Two teenage girls, one is smiling. | A girl is not smiling. | E | 3.62 |
| Group of men at a business meeting. | There are no women at the business meeting | E | 3.65 |
| A guy on a bike goes vertical near a ramp with a grassy, hilly terrain behind him. | A man is inside. | E | 3.51 |
| The sky appears clear. | There is nothing visible in the sky | E | 3.77 |
| | **TracIn Scores Head (easiest)** | | |
| A bicyclist rounds a turn, followed by a cameraman. | A man is driving a car around the roundabout | C | 0.00 |
| A young man on a bicycling is jumping up the stairs on his bike. | A man is eating lunch in the park. | C | 0.00 |
| The racing dog has a muzzle and is wearing striped jersey # 8. | The dog is sleep outside. | C | 0.00 |
| 2 men are sparing, 1 is in the process of taking down the other. | Nobody is sparing. | C | 0.00 |
| a tennis player hits the ball. | A tennis player is swimming across the ocean. | C | 0.00 |
| An old man is holding two ice cream cones as he walks through the park. | The person is sitting in the car. | C | 0.00 |
| A man with a big backpack is walking through a grass trail up a hill. | A woman with a big backpack is walking through a grass trail up a hill. | C | 0.00 |
| A man in a yellow jumpsuit is working. | Nobody is working. | C | 0.00 |
| A construction worker walks down the street, while others are at work. | A construction worker is sitting in his car. | C | 0.00 |
| A biker, wearing full protective gear including helmet, is jumping over a rock. | The biker is paralyzed in the hospital. | C | 0.00 |

Table 15: Shows the highest and lowest TracIn examples from SNLI.