# OpenReview forum: "Influence Scores at Scale for Efficient Language Data Sampling"
_EMNLP/2023/Conference — EMNLP 2023 Main_

### Official Review · Reviewer_CqZv · 2023-08-01

**Soundness:** 4

**Excitement:**

3: Ambivalent: It has merits (e.g., it reports state-of-the-art results, the idea is nice), but there are key weaknesses (e.g., it describes incremental work), and it can significantly benefit from another round of revision. However, I won't object to accepting it if my co-reviewers champion it.

**Paper Topic And Main Contributions:**

This paper presents a evaluation of influence scores for data reduction in language classification tasks. The authors benchmark five easy-to-implement influence scores on the Natural Language Inference (NLI) task, considering both noisy and clean data settings. Among the evaluated influence scores, they find that VoG performs best relative to the random-pruning baseline. Building upon these findings, the authors scale VoG to a real-world setting.


-------------Post-rebuttal---------------

I have read the author's response, and I think they have addressed my concerns. Therefore, I will improve my score.

**Reasons To Accept:**

- The paper is well-written and easy to follow.
- The authors provide extensive experimental results on influence scores, thereby contributing a valuable resource to the NLP community.

**Reasons To Reject:**

My primary concern revolves around the major conclusion of the paper, which appears to rely solely on the performance of BERT encoder on the SNLI dataset. This limited scope raises questions about the generalization of the findings to other datasets or tasks.

**Reproducibility:**

4: Could mostly reproduce the results, but there may be some variation because of sample variance or minor variations in their interpretation of the protocol or method.

**Reviewer Confidence:**

3: Pretty sure, but there's a chance I missed something. Although I have a good feel for this area in general, I did not carefully check the paper's details, e.g., the math, experimental design, or novelty.

---

> ### Author Rebuttal · Authors · 2023-08-29
>
> We thank you for your comments and feedback!
>
> A primary goal of our paper was to build upon existing influence-scoring literature in computer vision by applying influence scores to core NLU tasks for the first time. Hence we focus our evaluations on inherently different types of influence scores, for the purpose of selecting training data in a large-scale, real-life, language-based application. We believe this fills a gap in current influence-scoring literature, provides useful data points to practitioners, and can serve as a foundation for future experimentation.
>
> Our conclusions about the utility of influence scores for language data sampling were based on both SNLI evaluations as well as performance evaluations in a large-scale, online A/B user study, where downstream tasks included not only domain classification but also intent and slotting performance for a wide variety of NLU-related tasks. The scaled user study exposes the model to unconstrained human speech, which varies (often dramatically) in carrier phrase frequency, vocabulary, named entities distribution and other aspects from publicly available datasets such as SNLI. We dedicate a separate section to the real-life user study and showcase how influence-score based data selection can be effective even in an aggressive scenario of 50% reduction of training data, without loss of net accuracy.
>
> To the best of our knowledge, our work is the first to formally investigate influence scores at scale for language data and model architectures. We are excited about the potential for the research community to build upon our findings and extend experiments to new types of model architectures or language tasks. We will include specific recommendations in the paper discussion on how to expand usage into LLMs/decoder-only architectures to estimate influence of contextual examples.

---

### Official Review · Reviewer_ABYD · 2023-08-05

**Soundness:** 4

**Excitement:**

3: Ambivalent: It has merits (e.g., it reports state-of-the-art results, the idea is nice), but there are key weaknesses (e.g., it describes incremental work), and it can significantly benefit from another round of revision. However, I won't object to accepting it if my co-reviewers champion it.

**Paper Topic And Main Contributions:**

The main contributions of the paper are:

* Systematically benchmarked the applicability of a set of influence score techniques commonly applied in CV to an NLP task(SNLI).
* Benchmarking a subset of influence scores for data pruning on the SNLI dataset, and finding that out of the scores tested, VoG (variance of gradients) performed best compared to a random sampling baseline.
* Applying VoG to a NLU system in production and showing that training on a subset of data selected by the VoG scores did not lead to regression on key metrics.

**Questions For The Authors:**

A: Can we extend this technique to selecting which in-context examples are most likely to give the best performance?
B: is it possible to come up with some variants of influence scores to decide which parts of the pretraining corpus is important? the problem here  would be that we don’t have any labels.
C: Are these findings transferable to multilingual models?
D: How sensitive are influence scores to pretraining schemes and pretraining corpora?

**Reasons To Accept:**

* Clearly written and easy to read.
* Extensive set of experiments with a strong focus on using the techniques in production .
* Validates empirical results comprehensively for applying techniques to the system in production with both offline and online A/B tests.
* These influence score techniques would allow practitioners to identify the subset of the training data that is most important to performance and annotate more samples the belong to such categories.

**Reasons To Reject:**

No major reasons to reject, but I have some general concerns about the techniques discussed.

* This seems to solve the problem of deciding which examples are important in hindsight which concerns applicability of the technique to a system in production. When even small % differences in downstream performance can have material impact on company revenue, would they not prefer to train models on as much data as possible? Identifying which samples are easy or hard once they have been labelled can help determine which type of examples we need more annotations of, but the results showing regression on less common requests is concerning for people who make the decision to deploy the model into production.

**Reproducibility:**

3: Could reproduce the results with some difficulty. The settings of parameters are underspecified or subjectively determined; the training/evaluation data are not widely available.

**Reviewer Confidence:**

4: Quite sure. I tried to check the important points carefully. It's unlikely, though conceivable, that I missed something that should affect my ratings.

---

> ### Author Rebuttal · Authors · 2023-08-29
>
> We thank the reviewer for their comments and positive feedback!
>
> It would indeed be ideal to train on as much clean data as possible. However, problems arise not only due to raising training costs, but also because of mounting privacy and intellectual property concerns, customer data deletion requests, capacity constraints of human annotators, and governmental legislation that sets data retention policies (which can lead to very uneven data losses across classes when data expires), or legislation that defaults customers to opt-out of data submission (creating group biases in the available training data). Our goal was, in part, to understand how empirical data-scoring metrics could be used to surface, quantify, diagnose, and ultimately mitigate these risks. For example, given the risk of losing a certain percent of training data due to an data-expiration policy, it is paramount to develop empirical metrics that can surface the most critical and impactful utterances to expose annotators to.
>
> Second, we note that we chose a very aggressive target of a 50% reduction in training data in our online experiments. A regression on less common requests is indeed a cause for concern, but we aimed to highlight the efficacy of our method on overall metrics in a fairly extreme “proof of concept” case. In practical applications, we believe that our method would perform better via iterative instrumentation with gradual reductions and iterative scoring of training data, which is supported by results from the 14%-reduction experiment.
>
> Regarding your questions:
>
> A: We agree that applying this technique to in-context learning would be an exciting next step! We will include further discussion on implementing VoG scores contextually in LLMs/decoder-only architecture in our camera-ready version, but note that one potential approach would be to look at variance of model weights (e.g., in attention heads) in specific layers across the input sequence. This could offer an interpretable metric for pinpointing influential contextual examples, at the level of specific tokens (see, e.g., Grosse et al. [https://arxiv.org/abs/2308.03296](https://arxiv.org/abs/2308.03296) along similar lines). Given rising concerns over bias, toxicity, and fairness in language models , we believe there is a critical need for straightforward, cost-effective metrics that can estimate in-context influence. Our work develops the foundational understanding necessary to make progress on that problem by generalizing results from the computer vision field (such as those scores that approximate more computationally expensive influence functions) to language-based tasks at scale for the first time.
>
> B: Adapting these scores to model pretraining through a self-supervised setup would be an interesting line of investigation (for example, assessing the impact of removal of copyrighted pretraining data). One approach would be to compute an ensemble of the VoG score over all of the model outputs, rather than the output at a specific ground truth "label". However, applying this to an entire corpus of pretraining text would be computationally infeasible, so some degree of coarse-graining or clustering would be required to obtain representative samples from different categories (such as Books, Wikipedia, etc.) as estimators for the influence of that category.
>
> C: We believe these findings should generalize to multilingual setups, since the only ingredients are training artifacts (e.g., gradients, losses, etc.) that should, in principle, convey the relevant signal to determine data influence regardless of the specifics of the overall training data distribution. That said, most empirical influence scores do reflect some information about the model inductive bias and some component of the training-data distribution. For certain multilingual setups where training artifacts used to produce scores were derived from training data differing greatly from the to-be-scored data, the mismatch in these two sets of data may preclude or reduce the usefulness of the resulting scores for some downstream applications.
>
> D: While we cannot definitively answer without additional investigation, we can comment that it is challenging to assess the quality and nature of pretraining during finetuning (when we compute these scores). In this sense, we don't expect the efficacy of influence scores to be sensitive to the precise details of pretraining. However, there is a component of inductive bias reflected in these scores (for instance, in appendix B, we see a shift in accuracy and pruning curves resulting in the change in inductive bias coming from increasing model capacity.) It would be interesting to make this precise in future work — how much of dataset difficulty comes from pretraining and/or model inductive bias vs. information extracted about the data distribution?

---

### Official Review · Reviewer_g8MK · 2023-08-07

**Soundness:** 4

**Excitement:**

3: Ambivalent: It has merits (e.g., it reports state-of-the-art results, the idea is nice), but there are key weaknesses (e.g., it describes incremental work), and it can significantly benefit from another round of revision. However, I won't object to accepting it if my co-reviewers champion it.

**Paper Topic And Main Contributions:**

The paper explores the applicability of training dataset pruning via (five) influence scores in language classification tasks. It tries to find answers to several questions such as how useful the influence scores are for determining optimal data selection strategies, which scores work best, and evaluating these scores against a random sampling baseline in noisy and clean data settings.

**Reasons To Accept:**

The paper is generally well-written, and clear in terms of what it tries to investigate (problem formulation) and methodology. The study (on SNLI) looks decent and it provides some potentially useful insights to the community.

**Reasons To Reject:**

I am writing questions and reasons to reject together below:

1) In line 037-039: "A common challenge in fine-tuning transformer-based models is selecting which examples are most important for learning"---is it really a challenge anymore? Cheery picking examples might be infeasible when it comes to training larger models such as LLMs.

2) One of the concerns I have with this paper is its impact, the study has been carried out in a controlled setting (SNLI dataset, etc.) and thus inferences made may not generalize well for a wider range of problems (datasets and experimental settings).

**Reproducibility:**

3: Could reproduce the results with some difficulty. The settings of parameters are underspecified or subjectively determined; the training/evaluation data are not widely available.

**Reviewer Confidence:**

4: Quite sure. I tried to check the important points carefully. It's unlikely, though conceivable, that I missed something that should affect my ratings.

---

> ### Author Rebuttal · Authors · 2023-08-29
>
> We thank you for the review and constructive comments!
>
> Indeed, the standard approach to pre-training large language models involves training on all available data, which is simple though costly. However, there are formidable data-selection challenges for language models that influence scores can help to address, such as using scores to flag data most influential for various failure modes (e.g., toxic and biased response generation). Other practical application challenges that influence scores could be useful for include: assessing the influence of specific training data examples on the response stability with respect to prompt modifications; assessing the model-performance impact of copyrighted material in the training data; measuring the influence of sensitive private information present in big data lakes on model output; and quantifying the potential of model output surfacing private customer data that has been memorized by that model (we note this work by Grosse et al. along similar lines: [https://arxiv.org/abs/2308.03296](https://arxiv.org/abs/2308.03296)). We plan to explicitly incorporate these motivations in a paper revision, as well as a technical discussion on fruitful directions we see in applying influence scoring to LLMs.
>
> In addition, while LLMs are groundbreaking, many real-life NLU systems, including voice assistants and fast response/detection systems, require task understanding, latency and precision stability for which LLMs alone may not be an appropriate architectural choice. For example, low-resource languages may necessitate more classical model architectures, for which careful data selection becomes paramount. In other cases, standard LLM training approaches may be too costly or resource intensive, leading to the need for modeling approaches that are trained on specific, high-quality subsets of data (see, e.g., Gunasekar et al., [https://arxiv.org/abs/2306.11644](https://arxiv.org/abs/2306.11644)).
>
> We share your concern about the need for realistic evaluation of influence-scoring data selection approaches we experimented with. This is why we also conducted a large-scale user study evaluating our techniques in an uncontrolled speech online A/B setting. Statistical analysis on the results of this large-scale user study allowed us to empirically validate the impact of the explored influence-scoring techniques on real-life, unconstrained data that reflect practical challenges often not reflected in public datasets, such as noisy and evolving speech patterns, diverse vocabularies, dialects, carrier phrases, and named entity frequencies that deviate from what was provided to a model at training time. Even under these challenging evaluation conditions and under aggressive training-data reduction we still saw the utility of influence scores for data pruning. We will enhance the discussion to make this scaled user study more prominent as well.

---

### Meta-Review · Area_Chair_pqXA · 2023-09-17

**Recommendation:** 4

**Metareview:**

The paper focuses on the use of influence scores to select important examples for fine-tuning transformer-based models. There is a consensus on the clarity and readability of the paper. The paper is well-written and easy to follow. Reviewers also agree that the paper receives praise for its extensive set of experiments, which includes both offline and online A/B tests, and using the techniques in a production environment is a strong aspect of the study. This practical orientation is seen as valuable to the NLP community. Besides, the proposed influence score techniques are seen as potentially valuable for practitioners. They can help identify subsets of training data that are most important for model performance, facilitating better annotation strategies.

Although the reviewers do not have any major concerns, they question the relevance of the challenge mentioned in the paper's introduction, specifically regarding the selection of important examples for fine-tuning transformer-based models. They argue that this might not be a significant challenge anymore, particularly with larger models like LLMs. The paper should address this concern by providing more context and justification for this challenge. One reviewer expresses their concern about the limited scope of the study, as it focuses on a controlled setting using the SNLI dataset. They worry that the inferences drawn from this specific dataset may not generalize well to a wider range of problems, datasets, and experimental settings. The paper should acknowledge this limitation and discuss its implications.

In conclusion, the paper is commended for its clarity, experimental rigor, and potential utility for practitioners in the NLP community. However, it needs to address concerns related to the relevance of the challenge and the generalizability of its findings. Addressing these concerns and providing more context could strengthen the paper's overall contribution.

---

### Decision · Program_Chairs · 2023-10-07

**Decision:**

Accept-Main

**Comment:**

The paper focuses on the use of influence scores to select important examples for fine-tuning transformer-based models. There is a consensus on the clarity and readability of the paper. The paper is well-written and easy to follow. Reviewers also agree that the paper receives praise for its extensive set of experiments, which includes both offline and online A/B tests, and using the techniques in a production environment is a strong aspect of the study. This practical orientation is seen as valuable to the NLP community. Besides, the proposed influence score techniques are seen as potentially valuable for practitioners. They can help identify subsets of training data that are most important for model performance, facilitating better annotation strategies.

Although the reviewers do not have any major concerns, they question the relevance of the challenge mentioned in the paper's introduction, specifically regarding the selection of important examples for fine-tuning transformer-based models. They argue that this might not be a significant challenge anymore, particularly with larger models like LLMs. The paper should address this concern by providing more context and justification for this challenge. One reviewer expresses their concern about the limited scope of the study, as it focuses on a controlled setting using the SNLI dataset. They worry that the inferences drawn from this specific dataset may not generalize well to a wider range of problems, datasets, and experimental settings. The paper should acknowledge this limitation and discuss its implications.

In conclusion, the paper is commended for its clarity, experimental rigor, and potential utility for practitioners in the NLP community. However, it needs to address concerns related to the relevance of the challenge and the generalizability of its findings. Addressing these concerns and providing more context could strengthen the paper's overall contribution.